# REASONING ON GRAPHS: FAITHFUL AND INTERPRETABLE LARGE LANGUAGE MODEL REASONING

**Linhao Luo, Yuan-Fang Li, Gholamreza Haffari**
Monash University
Australia
{linhao.luo,yuanfang.li,Gholamreza.Haffari}@monash.edu

**Shirui Pan**[*]
Griffith University
Australia
s.pan@griffith.edu.au

## ABSTRACT

Large language models (LLMs) have demonstrated impressive reasoning abilities in complex tasks. However, they lack up-to-date knowledge and experience hallucinations during reasoning, which can lead to incorrect reasoning processes and diminish their performance and trustworthiness. Knowledge graphs (KGs), which capture vast amounts of facts in a structured format, offer a reliable source of knowledge for reasoning. Nevertheless, existing KG-based LLM reasoning methods only treat KGs as factual knowledge bases and overlook the importance of their structural information for reasoning. In this paper, we propose a novel method called reasoning on graphs (RoG) that synergizes LLMs with KGs to enable faithful and interpretable reasoning. Specifically, we present a planning-retrieval-reasoning framework, where RoG first generates relation paths grounded by KGs as faithful plans. These plans are then used to retrieve valid reasoning paths from the KGs for LLMs to conduct faithful reasoning. Furthermore, RoG not only distills knowledge from KGs to improve the reasoning ability of LLMs through training but also allows seamless integration with any arbitrary LLMs during inference. Extensive experiments on two benchmark KGQA datasets demonstrate that RoG achieves state-of-the-art performance on KG reasoning tasks and generates faithful and interpretable reasoning results[1].

## 1 INTRODUCTION

Large language models (LLMs) have shown great performance in many NLP tasks (Brown et al., 2020; Bang et al., 2023). What's especially striking is their ability to handle complex tasks through reasoning (Wei et al., 2022; Huang & Chang, 2023). To further unleash LLMs' reasoning ability, the plan-and-solve paradigm (Wang et al., 2023c) has been proposed, in which LLMs are prompted to generate a plan and execute each reasoning step. In this way, LLMs decompose complex reasoning tasks into a series of sub-tasks and solve them step by step (Khot et al., 2022).

Despite their success, LLMs are still limited by the lack of knowledge and prone to hallucinations during reasoning, which can lead to errors in reasoning processes (Hong et al., 2023; Wang et al., 2023b). For example, as shown in Figure 1, LLMs do not have the latest knowledge and hallucinate an incorrect reasoning step: "has a daughter". These issues largely diminish the performance and trustworthiness of LLMs in high-stakes scenarios, such as legal judgment and medical diagnosis.

To tackle the issues, knowledge graphs (KGs) have been incorporated to improve the reasoning ability of LLMs (Pan et al., 2024; Luo et al., 2023a). KGs capture abundant factual knowledge in a structured format, which provides a faithful knowledge source for reasoning. As a typical reasoning

---

[*]Corresponding author.
[1]Code and data are available at: https://github.com/RManLuo/reasoning-on-graphs

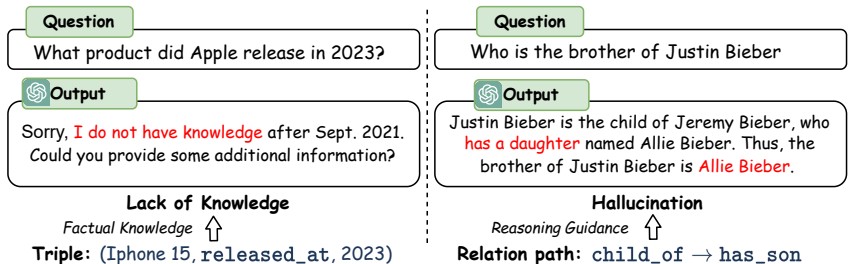

Figure 1: The issues of lack of knowledge and hallucination in LLMs reasoning and how they can be addressed by triples and relation paths from KGs.

task, knowledge graph question answering (KGQA) aims to obtain answers based on knowledge from KGs (Sun et al., 2019). Previous works that jointly use KGs and LLMs for KGQA reasoning can be broadly divided into two categories: 1) semantic parsing methods (Lan & Jiang, 2020; Ye et al., 2022), which use LLMs to convert questions into logical queries that are executed on KGs to obtain answers; and 2) retrieval-augmented methods (Li et al., 2023; Jiang et al., 2023), which retrieve triples from KGs as knowledge context and uses LLMs to obtain the final answers.

Although semantic parsing methods can generate more accurate and interpretable results by leveraging reasoning on KGs, the generated logical queries can often be non-executable and yield no answers, due to syntax and semantic limitations (Yu et al., 2022a). Retrieval-augmented methods are more flexible and exploit the ability of LLMs for reasoning. However, they only treat KGs as factual knowledge bases and overlook the importance of their structural information for reasoning (Jiang et al., 2022). For instance, as shown in Figure 1, a *relation path*, which is a sequence of relations, "child_of→has_son" can be used to obtain answers to the question "Who is the brother of Justin Bieber?". Therefore, it is essential to enable LLMs to directly reason on KGs to achieve faithful and interpretable reasoning.

In this paper, we propose a novel method called reasoning on graphs (RoG) that synergizes LLMs with KGs to conduct faithful and interpretable reasoning. To alleviate the issues of hallucinations and lack of knowledge, we present a *planning-retrieval-reasoning* framework, where RoG first generates relation paths grounded by KGs as faithful plans via the *planning module*. These plans are then used to retrieve valid reasoning paths from KGs to conduct faithful reasoning by the *retrieval-reasoning module*. In this way, we not only retrieve the latest knowledge from KGs but also consider the guidance of KG structure for reasoning and explanations. Moreover, the planning module of RoG can be plug-and-play with different LLMs during inference to improve their performance. Based on this framework, RoG is optimized by two tasks: 1) *planning optimization*, where we distill knowledge from KGs into LLMs to generate faithful relation paths as plans; and 2) *retrieval-reasoning optimization*, where we enable LLMs to conduct faithful reasoning based on retrieved paths and generate interpretable results. We conduct extensive experiments on two benchmark KGQA datasets, and the results demonstrate that RoG achieves state-of-the-art performance on KG reasoning tasks and generates faithful and interpretable reasoning results.

## 2 RELATED WORK

**LLM Reasoning Prompt.** Many studies have been proposed to harness the reasoning ability of LLMs to handle complex tasks through prompting (Wei et al., 2022; Wang et al., 2022; Yao et al., 2023; Besta et al., 2023). Plan-and-solve (Wang et al., 2023c) prompts LLMs to generate a plan and conduct reasoning based on it. DecomP (He et al., 2021) prompts LLMs to decompose the reasoning task into a series of sub-tasks and solve them step by step. However, the problem of hallucinations and lack of knowledge affect the faithfulness of LLMs' reasoning. ReACT (Yao et al., 2022) treats LLMs as agents, which interact with the environment to get the latest knowledge for reasoning. To explore faithful reasoning, FAME (Hong et al., 2023) introduces the Monte-Carlo planning to generate faithful reasoning steps. RR (He et al., 2022) and KD-CoT Wang et al. (2023b) further retrieve relevant knowledge from KGs to produce faithful reasoning plans for LLMs.

**Knowledge Graph Question Answering (KGQA).** Conventional *embedding-based methods* represent the entities and relations in embedding space and design special model architectures (e.g., Key-Value memory networks, sequential models, and graph neural networks) to reason answers

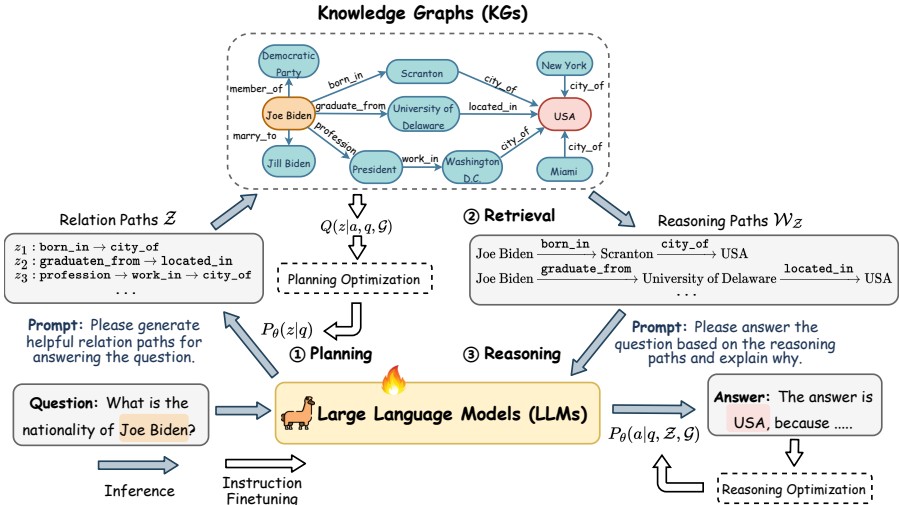

Figure 2: The overall framework of reasoning on graphs (RoG). 1) given a question, we first prompt LLMs to generate several relation paths that are grounded by KGs as plans. 2) Then, we retrieve reasoning paths from KGs using the plans. 3) Finally, we conduct faithful reasoning based on the retrieved reasoning paths and generate answers with interpretable explanations. The orange and red rectangles denote the entities mentioned in the question and answer, respectively.

(Miller et al., 2016; He et al., 2021; Yasunaga et al., 2021). To integrate LLMs for KGQA, *retrieval-augmented methods* aim to retrieve the relative facts from the KGs to improve the reasoning performance (Li et al., 2023; Karpukhin et al., 2020). Recently, UniKGQA (Jiang et al., 2022) which unifies the graph retrieval and reasoning process into a single model with LLMs, achieves STOA performance. *Semantic parsing methods* convert the question into a structural query (e.g., SPARQL) by LLMs, which can be executed by a query engine to reason the answers on KGs (Sun et al., 2020; Lan & Jiang, 2020). However, these methods heavily rely on the quality of generated queries. If the query is not executable, no answers will be generated. DECAF (Yu et al., 2022a) combines semantic parsing and LLMs reasoning to jointly generate answers, which also reach salient performance on KGQA tasks.

## 3 PRELIMINARY

**Knowledge Graphs (KGs)** contain abundant factual knowledge in the form of a set of triples: $\mathcal{G} = \{(e, r, e')|e, e' \in \mathcal{E}, r \in \mathcal{R}\}$, where $\mathcal{E}$ and $\mathcal{R}$ denote the set of entities and relations, respectively.

**Relation Paths** are a sequence of relations: $z = \{r_1, r_2, \ldots, r_l\}$, where $r_i \in \mathcal{R}$ denotes the $i$-th relation in the path and $l$ denotes the length of the path.

**Reasoning Paths** are the instances of a relation path $z$ in KGs: $w_z = e_0 \xrightarrow{r_1} e_1 \xrightarrow{r_2} \ldots \xrightarrow{r_l} e_l$, where $e_i \in \mathcal{E}$ denotes the $i$-th entity and $r_i$ denotes the $i$-th relation in the relation path $z$.

**Example 1.** Given a relation path: $z = \mathtt{marry\_to} \rightarrow \mathtt{father\_of}$, a reasoning path instance could be: $w_z = \text{Alice} \xrightarrow{\mathtt{marry\_to}} \text{Bob} \xrightarrow{\mathtt{father\_of}} \text{Charlie}$, which denotes "Alice" is married to "Bob" and "Bob" is the father of "Charlie".

**Knowledge Graph Question Answering (KGQA)** is a typical reasoning task based on KGs. Given a natural language question $q$ and a KG $\mathcal{G}$, the task aims to design a function $f$ to predict answers $a \in \mathcal{A}_q$ based on knowledge from $\mathcal{G}$, i.e., $a = f(q, \mathcal{G})$. Following previous works (Sun et al., 2019; Jiang et al., 2022), we assume the entities $e_q \in \mathcal{T}_q$ mentioned in $q$ and answers $a \in \mathcal{A}_q$ are labeled and linked to the corresponding entities in $\mathcal{G}$, i.e., $\mathcal{T}_q, \mathcal{A}_q \subseteq \mathcal{E}$.

## 4 APPROACH

In this section, we introduce our method: reasoning on graphs (RoG), which contains two components: 1) a *planning* module that generates relation paths grounded by KGs as faithful plans; 2)

a *retrieval-reasoning* module that first retrieves valid reasoning paths from KGs according to the plans, then conducts faithful reasoning based on retrieved reasoning paths and generates answers with interpretable explanations. The overall framework of RoG is illustrated in Figure 2.

## 4.1 REASONING ON GRAPHS: PLANNING-RETRIEVAL-REASONING

Recently, many techniques have been explored to improve the reasoning ability of LLMs by planning, which first prompts LLMs to generate a reasoning plan and then conduct reasoning based on it (Wang et al., 2023c). However, LLMs are known for having hallucination issues, which are prone to generating incorrect plans and leading to wrong answers (Ji et al., 2023). To address this issue, we present a novel *planning-retrieval-reasoning* framework, which makes the reasoning plans grounded by KGs and then retrieves faithful reasoning paths for LLM reasoning.

Relation paths, which capture semantic relations between entities, have been utilized in many reasoning tasks on KGs (Wang et al., 2021; Xu et al., 2022). Moreover, compared to the dynamically updated entities, the relations in KGs are more stable (Wang et al., 2023a). By using relation paths, we can always retrieve the latest knowledge from KGs for reasoning. Therefore, relation paths can serve as faithful plans for reasoning the answer to KGQA task.

**Example 2.** Given a question "Who is the child of Alice", we can generate a relation path as the plan: $z = \texttt{marry\_to} \rightarrow \texttt{father\_of}$. This relation path expresses the plan: 1) find the person that "Alice" is married to; 2) find the child of that person. We can execute the plan (relation path) by retrieving a reasoning path from KGs as: $w_z = \text{Alice} \xrightarrow{\texttt{marry\_to}} \text{Bob} \xrightarrow{\texttt{father\_of}} \text{Charlie}$. Finally, we can answer the question based on the reasoning path, which is "Charlie".

By treating relation paths as plans, we can make sure the plans are grounded by KGs, which enables LLMs to conduct faithful and interpretable reasoning on graphs. In a nutshell, we formulate our RoG as an optimization problem that aims to maximize the probability of reasoning the answer from a knowledge graph $\mathcal{G}$ w.r.t the question $q$ by generating relation paths $z$ as the plan:

$$P_\theta(a|q, \mathcal{G}) = \sum_{z \in \mathcal{Z}} P_\theta(a|q, z, \mathcal{G}) P_\theta(z|q), \tag{1}$$

where $\theta$ denotes the parameters of LLMs, $z$ denotes the relation paths (plans) generated by LLMs, and $\mathcal{Z}$ denotes the set of possible relation paths. The latter term $P_\theta(z|q)$ is the probability of generating a faithful relation path $z$ grounded by KG, given the question $q$, which is realized by the *planning* module. The former term $P_\theta(a|q, z, \mathcal{G})$ is the probability of reasoning an answer $a$ given the question $q$, relation path $z$, and KG $\mathcal{G}$, which is computed by the *retrieval-reasoning* module.

## 4.2 OPTIMIZATION FRAMEWORK

Despite the advantage of generating relation paths as plans, the LLMs have zero knowledge of the relations contained in KGs. Therefore, LLMs cannot directly generate relation paths grounded by KGs as faithful plans. Moreover, LLMs might not understand the reasoning paths correctly and conduct reasoning based on them. To address these issues, we design two instruction tuning tasks: 1) *planning optimization*, which distills the knowledge from KGs into LLMs to generate faithful relation paths as plans; 2) *retrieval-reasoning optimization*, which enables LLMs to reason based on the retrieved reasoning paths.

The objective function in equation 1 can be optimized by maximizing the evidence lower bound (ELBO) (Jordan et al., 1999), which is formulated as

$$\log P(a|q, \mathcal{G}) \geq \mathbb{E}_{z \sim Q(z)}[\log P_\theta(a|q, z, \mathcal{G})] - D_{\mathrm{KL}}(Q(z) \| P_\theta(z|q)), \tag{2}$$

where $Q(z)$ denotes the posterior distribution of faithful relation paths grounded by KGs. The latter term minimizes the KL divergence between the posterior and the prior, which encourages LLMs to generate faithful relation paths (planning optimization). The former term maximizes the expectation that retrieval-reasoning module generates correct answers based on the relation paths and KGs (retrieval-reasoning optimization).

**Planning optimization.** In planning optimization, we aim to distill the knowledge from KGs into LLMs to generate faithful relation paths as plans. This can be achieved by minimizing the KL divergence with the posterior distribution of faithful relation paths $Q(z)$, which can be approximated by the valid relation paths in KGs.

Given a question $q$ and answer $a$, we could find the path instances $w_z(e_q, e_a) = e_q \xrightarrow{r_1} e_1 \xrightarrow{r_2} \ldots \xrightarrow{r_l} e_a$ connecting $e_q$ and $e_a$ in KGs. The corresponding relation path $z = \{r_1, r_2, \ldots, r_l\}$ can be considered valid and serve as a faithful plan for answering the question $q$. The posterior distribution $Q(z)$ can be formally approximated as

$$Q(z) \simeq Q(z|a, q, \mathcal{G}) = \begin{cases} \dfrac{1}{|\mathcal{Z}|}, \exists w_z(e_q, e_a) \in \mathcal{G}, \\ 0, else, \end{cases} \tag{3}$$

where we assume a uniform distribution over all valid relation paths $\mathcal{Z}$, and $\exists w_z(e_q, e_a) \in \mathcal{G}$ denote the existence of a path instance connecting the question $e_q$ and answer $e_a$ entities in $\mathcal{G}$. Therefore, the KL divergence can be calculated as

$$\mathcal{L}_{\text{plan}} = D_{\text{KL}}(Q(z)\|P_\theta(z|q)) = D_{\text{KL}}(Q(z|a, q, \mathcal{G})\|P_\theta(z|q)) \simeq -\frac{1}{|\mathcal{Z}^*|} \sum_{z \in \mathcal{Z}^*} \log P_\theta(z|q), \tag{4}$$

where we use the shortest paths $\mathcal{Z}^* \subseteq \mathcal{Z}$ between $e_q$ and $e_a$ in KGs as supervision signals (Zhang et al., 2022). The detailed derivation of can be found in Appendix A.1. By optimizing the equation 4, we maximize the probability of LLMs generating faithful relation paths through distilling the knowledge from KGs.

**Retrieval-reasoning optimization.** In retrieval-reasoning optimization, we aim to enable LLMs to conduct reasoning based on the retrieved reasoning paths. For the retrieval-reasoning module, we follow the FiD framework (Izacard & Grave, 2021; Singh et al., 2021), which allows reasoning on multiple retrieved reasoning paths[2], formulated as

$$P_\theta(a|q, \mathcal{Z}, \mathcal{G}) = \prod_{z \in \mathcal{Z}} P_\theta(a|q, z, \mathcal{G}). \tag{5}$$

By approximating the expectation with K sampled plans $\mathcal{Z}_K^* \subseteq Z^*$, the objective function of reasoning optimization can be written as

$$\mathcal{L}_{\text{reason}} = \mathbb{E}_{z \sim Q(z|a, q, \mathcal{G})}[\log P_\theta(a|q, z, \mathcal{G})] = \sum_{z \in \mathcal{Z}_K^*} \log P_\theta(a|q, z, \mathcal{G}) = \log P_\theta(a|q, \mathcal{Z}_K^*, \mathcal{G}). \tag{6}$$

This maximizes the probability of LLMs generating correct answers based on the retrieved reasoning paths.

The final objective function of RoG is the combination of the planning optimization and retrieval-reasoning optimization, which can be formulated as

$$\mathcal{L} = \log \underbrace{P_\theta(a|q, \mathcal{Z}_K^*, \mathcal{G})}_{\text{Retrieval-reasoning}} + \underbrace{\frac{1}{|\mathcal{Z}^*|} \sum_{z \in Z^*} \log P_\theta(z|q)}_{\text{Planning}}. \tag{7}$$

From equation 7, we can see that we adopt the same LLM for both planning and reasoning, which are jointly trained on two instruction-tuning tasks, i.e., (planning and retrieval-reasoning). We will discuss the implementation details of these two tasks in the following subsections.

## 4.3 PLANNING MODULE

The planning module aims to generate faithful relation paths as plans for answering the question. To utilize the instruction-following ability of LLMs (Wei et al., 2021), we design a simple instruction template that prompts LLMs to generate relation paths:

> Please generate a valid relation path that can be helpful for answering the following question: `<Question>`

where `<Question>` indicates the question $q$. The question together with the instruction template is fed into LLMs to generate the relation paths, which are structurally formatted as a sentence:

---

[2]The FiD framework assumes each $z \in \mathcal{Z}$ contributes independently, where the probability $P_\theta(a|q, \mathcal{Z}, \mathcal{G})$ can be approximated as the product of each $P_\theta(a|q, z, \mathcal{G})$.

$$z = \texttt{<PATH>} \ r_1 \ \texttt{<SEP>} \ r_2 \ \texttt{<SEP>} \ \ldots \ \texttt{<SEP>} \ r_l \ \texttt{</PATH>}$$

where $\texttt{<PATH>}, \texttt{<SEP>}, \texttt{</PATH>}$ are special tokens indicating the start, separator, and end of the relation path, respectively[3].

Therefore, the optimization of $\mathcal{L}_{\text{plan}}$ can be achieved as

$$\arg\max_\theta \frac{1}{|\mathcal{Z}^*|} \sum_{z \in \mathcal{Z}^*} \log P_\theta(z|q) = \frac{1}{|\mathcal{Z}^*|} \sum_{z \in \mathcal{Z}^*} \log \prod_{i=1}^{|z|} P_\theta(r_i|r_{<i}, q), \tag{8}$$

where $P_\theta(z|q)$ denotes the prior distribution of generating faithful relation path $z$, and $P_\theta(r_i|r_{<i}, q)$ denotes the probability of each token in $z$ generated by LLMs.

## 4.4 RETRIEVAL-REASONING MODULE

**Retrieval**. Given a question $q$ and a relation path as plan $z$, the retrieval module aims to retrieve the reasoning paths $w_z$ from KG $\mathcal{G}$. The retrieval process can be conducted by finding paths in $\mathcal{G}$ that start from the question entities $e_q$ and follow the relation paths $z$, formulated as

$$\mathcal{W}_z = \{w_z(e_q, e_*)|w_z(e_q, e_*) = e_q \xrightarrow{r_1} e_1 \xrightarrow{r_2} \ldots \xrightarrow{r_l} e_{a*}, w_z(e_q, e_*) \in \mathcal{G}\}. \tag{9}$$

We adopt a constrained breadth-first search to retrieve the reasoning paths $w_z$ from KGs. In experiments, all retrieved paths are used for reasoning. The detailed retrieval algorithm can be found in Appendix A.3.

Despite we can utilize the retrieved reasoning paths and directly get the answers with majority voting. The retrieved reasoning paths could be noisy and irrelevant to the questions, which leads to incorrect answers (He et al., 2021; Zhang et al., 2022). Therefore, we propose a reasoning module to explore the ability of LLMs to identify the important reasoning paths and answer the questions based on them.

**Reasoning**. The reasoning module takes the question $q$ and a set of reasoning paths $\mathcal{W}_z$ to generate answers $a$. Similarly, we design a reasoning instruction prompt to guide LLMs to conduct reasoning based on the retrieved reasoning paths $\mathcal{W}_z$. The $\mathcal{W}_z$ are also formatted as a series of structural sentences. The detailed prompt can be found in Appendix A.10.

The optimization of $\mathcal{L}_{\text{reason}}$ can be written as

$$\arg\max_\theta \log P_\theta(a|q, \mathcal{Z}_K^*, \mathcal{G}) = \log \sum_{z \in \mathcal{Z}_K^*} \sum_{w_z \in \mathcal{W}_z} \prod_{i=1}^{|a|} P_\theta(t_i|t_{<i}, q, w_z), \tag{10}$$

where $P_\theta(a|q, \mathcal{Z}_K, \mathcal{G})$ denotes probability of reasoning the correct answer $a$ based on $K$ relation paths $\mathcal{Z}_K$, and $t_*$ denote the tokens of answers $a$.

## 5 EXPERIMENT

In our experiments, we aim to answer the following research questions: **RQ1:** Can RoG achieve state-of-the-art performance on the KGQA tasks? **RQ2:** Can the planning module of RoG be integrated with other LLMs to improve their performance? **RQ3:** Can RoG conduct faithful reasoning and generate interpretable reasoning results?

### 5.1 EXPERIMENT SETTINGS

**Datasets.** We evaluate the reasoning ability of RoG on two benchmark KGQA datasets: WebQuestionSP (WebQSP) (Yih et al., 2016) and Complex WebQuestions (CWQ) (Talmor & Berant, 2018), which contain up to 4-hop questions. Freebase (Bollacker et al., 2008) is the background knowledge graph for both datasets, which contains around 88 million entities, 20 thousand relations, and 126 million triples. The details of the datasets are described in Appendix A.4.

---

[3]The relation name $r_*$ could be split into multiple tokens. For example, "$\texttt{born\_in}$" could be split into "$\texttt{born}$" and "$\texttt{\_in}$" by tokenizer. In this way, we could fully utilize the semantic information in relation names and generalize to different KGs.

Table 1: Performance comparison with different baselines on the two KGQA datasets.

| Type | Methods | WebQSP | | CWQ | |
|---|---|---|---|---|---|
| | | Hits@1 | F1 | Hits@1 | F1 |
| Embedding | KV-Mem (Miller et al., 2016) | 46.7 | 34.5 | 18.4 | 15.7 |
| | EmbedKGQA (Saxena et al., 2020) | 66.6 | - | 45.9 | - |
| | NSM (He et al., 2021) | 68.7 | 62.8 | 47.6 | 42.4 |
| | TransferNet (Shi et al., 2021) | 71.4 | - | 48.6 | - |
| | KGT5 Saxena et al. (2022) | 56.1 | - | 36.5 | - |
| Retrieval | GraftNet (Sun et al., 2018) | 66.4 | 60.4 | 36.8 | 32.7 |
| | PullNet (Sun et al., 2019) | 68.1 | - | 45.9 | - |
| | SR+NSM (Zhang et al., 2022) | 68.9 | 64.1 | 50.2 | 47.1 |
| | SR+NSM+E2E (Zhang et al., 2022) | 69.5 | 64.1 | 49.3 | 46.3 |
| Semantic Parsing | SPARQL (Sun et al., 2020) | - | - | 31.6 | - |
| | QGG (Lan & Jiang, 2020) | 73.0 | 73.8 | 36.9 | 37.4 |
| | ArcaneQA (Gu & Su, 2022) | - | 75.3 | - | - |
| | RnG-KBQA (Ye et al., 2022) | - | 76.2 | - | - |
| LLMs | Flan-T5-xl (Chung et al., 2022) | 31.0 | - | 14.7 | - |
| | Alpaca-7B (Taori et al., 2023) | 51.8 | - | 27.4 | - |
| | LLaMA2-Chat-7B (Touvron et al., 2023) | 64.4 | - | 34.6 | - |
| | ChatGPT | 66.8 | - | 39.9 | - |
| | ChatGPT+CoT | 75.6 | - | 48.9 | - |
| LLMs+KGs | KD-CoT (Wang et al., 2023b) | 68.6 | 52.5 | 55.7 | - |
| | UniKGQA (Jiang et al., 2022) | 77.2 | 72.2 | 51.2 | 49.1 |
| | DECAF (DPR+FiD-3B) (Yu et al., 2022a) | 82.1 | **78.8** | - | - |
| | RoG | **85.7** | 70.8 | **62.6** | **56.2** |

**Baselines.** We compare `RoG` with 21 baselines grouping into 5 categories: 1) Embedding-based methods, 2) Retrieval-augmented methods, 3) Semantic parsing methods, 4) LLMs, and 5) LLMs+KGs methods. The details of each baseline are described in Appendix A.5.

**Evaluation Metrics.** Following previous works, we use Hits@1 and F1 as the evaluation metrics. Hits@1 measures the proportion of questions whose top-1 predicted answer is correct. Since a question may correspond to multiple answers, F1 considers the coverage of all answers, which balances the precision and recall of the predicted answers.

**Implementations.** For `RoG`, we use LLaMA2-Chat-7B (Touvron et al., 2023) as the LLM backbone, which is instruction finetuned on the training split of WebQSP and CWQ as well as Freebase for 3 epochs. We generate the top-3 relation paths using beam-search for each question. Since UniKGQA (Jiang et al., 2022) and DECAF (Yu et al., 2022a) are state-of-the-art methods, we directly refer their results and those of the other baselines reported in their papers for comparisons. For LLMs, we use zero-shot prompting to conduct KGQA. The detailed settings are described in Appendix A.6.

## 5.2 RQ1: KGQA PERFORMANCE COMPARISON

**Main Results.** In this section, we compare `RoG` with other baselines on KGQA tasks. The results are shown in Table 1. Our method achieves the best performance on both datasets across most metrics. Specifically, compared to the SOTA method DECAF (Yu et al., 2022a) on WebQSP, our method improves Hits@1 by 4.4%. On the CWQ dataset, which is more challenging due to multi-hop questions, our method improves both Hits@1 and F1 by 22.3% and 14.4% against the SOTA model UniKGQA (Jiang et al., 2022). These results demonstrate the superior reasoning ability of our method in KGQA.

Among other methods, retrieval-augmented approaches outperform conventional embedding-based methods by retrieving relevant subgraphs from KGs, which reduces reasoning complexity. Furthermore, SR+NSM and SR+NSM+E2E adopt relation paths-based retrieval which achieves better performance, highlighting the importance of relation paths. Semantic parsing methods perform better than retrieval methods on WebQSP but worse on CWQ due to the complexity of generating logical queries for complex questions in CWQ. Although LLMs-based methods achieve comparable performance, they are limited by hallucinations and lack of knowledge as shown in Section 5.4. The LLMs+KGs methods achieve the second-best performance, which demonstrates the effectiveness of unifying KGs and LLMs for reasoning.

**Ablation Study.** We conduct an ablation study to analyze the effectiveness of the *planning module* and *reasoning module* in our method (`RoG`). We compare four variants: 1) *w/o planning*, where we remove the planning module and perform reasoning without retrieved reasoning paths; 2) *w/o reasoning*, where we remove the reasoning module and use all answers from retrieved reasoning

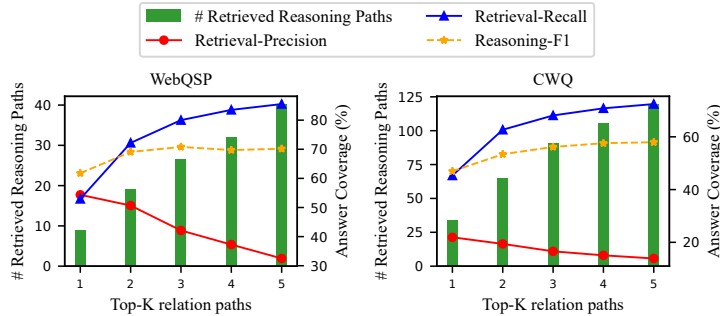

Figure 3: Faithfulness of top-$K$ generated relation paths. *Green bars* denote the average number of retrieved reasoning paths, *solid lines* denote the answer coverage of the retrieved paths, and *dashed line* denote the answer coverage of the reasoning module based on the retrieved reasoning paths.

paths as results; 3) *w/ random plans*, where we randomly retrieve reasoning paths from KGs and feed them into the reasoning module; 4) *w/ vote reasoning*, where we adopt the majority voting to select top-5 answers from retrieved reasoning paths. The results are shown in Table 2.

From the results, it is evident that without a planning module, our method degenerates to conventional LLMs that solely rely on questions as input, suffering from the lack of knowledge issue. Although removing the reasoning module leads to high recall due to an increased number of

Table 2: Ablation studies of RoG.

| Method | WebQSP | | | CWQ | | |
|---|---|---|---|---|---|---|
| | Precision | Recall | F1 | Precision | Recall | F1 |
| RoG | **74.77** | 75.84 | **70.81** | **57.69** | 58.19 | **56.17** |
| RoG w/o planning | 57.26 | 50.16 | 49.69 | 35.35 | 34.77 | 33.76 |
| RoG w/o reasoning | 46.90 | **79.85** | 49.56 | 18.88 | **67.89** | 22.26 |
| RoG w/ random plans | 38.66 | 38.31 | 35.24 | 38.99 | 39.29 | 37.64 |
| RoG w/ vote reasoning | 54.80 | 60.44 | 47.96 | 22.92 | 47.98 | 26.52 |

answers, precision drops significantly because of noise in retrieved paths. This demonstrates the effectiveness of the reasoning module in identifying important reasoning paths and filtering out noise. Furthermore, using random plans achieves worse performance than removing the planning module, highlighting the importance of a planning module who generates faithful reasoning plans. Using a simple majority vote reasoning can improve the results which also demonstrate the necessity of reasoning module.

## 5.3 RQ2: PLUG-AND-PLAY RoG PLANNING MODULE

In this section, we evaluate the effectiveness of integrating the planning module of RoG with different LLMs during inference to improve their performance. Specifically, we first adopt the planning module of RoG to generate relation paths and feed the retrieved reasoning paths as context into different LLMs for reasoning. The results are presented in Table 3. To account for the fact that it is difficult to extract the number of answers from LLM's output. We only report the Hits@1 and Recall metrics.

Table 3: Effects of integrating the planning module of RoG with different LLMs for reasoning.

| Methods | WebQSP | | CWQ | |
|---|---|---|---|---|
| | Hits@1 | Recall | Hits@1 | Recall |
| ChatGPT | 66.77 | 49.27 | 39.90 | 35.07 |
| ChatGPT + RoG Planning | **81.51** | **71.60** | **52.68** | **48.51** |
| Alpaca-7B | 51.78 | 33.65 | 27.44 | 23.62 |
| Alpaca-7B + RoG Planning | **56.16** | **74.20** | **44.04** | **38.46** |
| LLaMA2-Chat-7B | 64.37 | 44.61 | 34.60 | 29.91 |
| LLaMA2-Chat-7B + RoG Planning | **74.20** | **56.16** | **56.41** | **51.99** |
| Flan-T5-xl | 30.95 | 17.08 | 14.69 | 12.25 |
| Flan-T5-xl + RoG Planning | **67.87** | **44.93** | **37.81** | **32.57** |

From the results, we can notice that the performance of all LLMs is substantially improved by integrating the planning module of RoG. Specifically, the Hits@1 of ChatGPT, Alpaca, LLaMA2, and Flan-T5 are improved by 8.5%, 15.3%, and 119.3%, respectively. This demonstrates that the planning module of RoG can be seamlessly integrated with other LLMs to improve their performance without retraining.

## 5.4 RQ3: FAITHFUL REASONING AND INTERPRETABLE RESULTS

**Quantitative Results.** To evaluate the faithfulness of the relation paths, we illustrate the quantitative results in Figure 3. In experiments, we sweep the number of top-$K$ relation paths generated by RoG. From results, we can see that the number of retrieved reasoning paths increases with $K$, which also leads to more answers to be covered (recall). This demonstrates the faithfulness of the

Table 4: Example of *lack of knowledge* for ChatGPT-CoT and interpretable predictions of RoG.

| Question | What type of government is used in the country with Northern District? |
|---|---|
| Answer | Parliamentary system |
| ChatGPT + CoT | To determine what type of government is used in a country with a Northern District, we need more specific information. The term "Northern District" does not specify the name of any particular country or provide enough context to identify its government type. Could you please provide additional details or clarify the specific country you are referring to? |
| Reasoning Paths | Northern District → location.administrative_division.first_level_division_of → Israel → government.form_of_government.countries → **Parliamentary system** |
| RoG | The type of government used in the country with the Northern District is a **Parliamentary system**.

Explanations:

1. Northern District → location.administrative_division.first_level_division_of → Israel: The Northern District is a part of Israel.
2. Israel → government.form_of_government.countries → Parliamentary system: Israel has a Parliamentary system of government.

So, by following the reasoning paths provided, we can conclude that the country with the Northern District, which is Israel, uses a Parliamentary system of government. |

Table 5: Example of the *hallucination* of ChatGPT-CoT and faithful reasoning of RoG.

| Question | Where is the home stadium of the team who won the 1946 World Series championship? |
|---|---|
| Answer | Busch Stadium |
| ChatGPT + CoT | Step 1: Identify the team that won the 1946 World Series championship. The team that won the 1946 World Series championship was the St. Louis Cardinals.

Step 2: Determine the home stadium of the St. Louis Cardinals during that time. The St. Louis Cardinals' home stadium during the 1946 season was Sportsman's Park.

Therefore, the possible answer to the question "Where is the home stadium of the team who won the 1946 World Series championship?" is: - Sportsman's Park. |
| Reasoning Paths | 1946 World Series → sports.sports_team.championships → St. Louis Cardinals → sports.sports_team.arena_stadium → **Busch Stadium**
1946 World Series → sports.sports_team.championships → St. Louis Cardinals → sports.sports_team.arena_stadium → Roger Dean Stadium |
| RoG | Based on the reasoning paths provided, the home stadium of the team who won the 1946 World Series championship is **Busch Stadium**. This conclusion is based on the following path:

**1. 1946 World Series → sports.sports_team.championships → St. Louis Cardinals → sports.sports_team.arena_stadium → Busch Stadium**

The reasoning path leads us to the St. Louis Cardinals as the team that won the 1946 World Series, and Busch Stadium is the stadium associated with the St. Louis Cardinals. Therefore, **Busch Stadium** is the home stadium of the team that won the 1946 World Series championship. |

relation paths by retrieving the answers. However, more retrieved reasoning paths also lead to more noise and retrieval time (shown in Appendix A.7.4), which decreases the precision and makes little contribution to the final results (reasoning-f1). Therefore, we set $K = 3$ in experiments.

**Case studies.** We also illustrate two case studies in Table 4 and Table 5. In Table 4, we can find that ChatGPT+CoT suffers from the lack of knowledge issue and cannot answer the question. On the contrary, RoG can generate faithful relation paths and retrieve valid reasoning paths from KGs for reasoning. Besides, RoG can provide interpretable explanations based on the reasoning paths. In Table 5, we can see that ChatGPT+CoT suffers from hallucinations and generates incorrect answers. In contrast, although the retrieved reasoning paths contain noises, the reasoning module can identify the correct reasoning paths and conduct faithful reasoning. These results demonstrate the effectiveness of RoG in conducting faithful reasoning and generating interpretable results. More cases can be found in Appendices A.8 and A.9.

## 6 CONCLUSION

In this paper, we propose a novel method called reasoning on graphs (RoG) that synergizes LLMs with KGs to conduct faithful and interpretable reasoning. To alleviate the issues of hallucinations and lack of knowledge, we present a planning-retrieval-reasoning framework, which allows LLMs to access the latest knowledge while reasoning based on faithful plans on graphs. RoG not only enhances the reasoning capability of LLMs by distilling knowledge from KGs through training but also enables seamless integration with any LLMs during inference. Extensive experiments on two benchmark KGQA datasets demonstrate the superiority of RoG in reasoning ability and interpretability.

## ACKNOWLEDGMENTS

This material is based on research partially sponsored by the DARPA Assured Neuro Symbolic Learning and Reasoning (ANSR) program under award number FA8750-23-2-1016 and the DARPA Knowledge Management at Scale and Speed (KMASS) program under award number HR00112220047. This research is partly supported by the ARC Future Fellowship FT190100039 to G.H. S. Pan was supported in part by the Australian Research Council (ARC) under grants FT210100097 and DP240101547, and the CSIRO – National Science Foundation (US) AI Research Collaboration Program.

## ETHICS STATEMENT

Our work is solely dedicated to tackling scientific problems and does not involve any human subjects, animals, or environmentally sensitive materials. Consequently, we do not anticipate any potential ethical risks or conflicts of interest in our study. Our research strives to adhere to the highest standards of scientific integrity and ethical conduct throughout the entire process, guaranteeing the validity and reliability of our findings.

## REPRODUCIBILITY STATEMENT

Our model has been meticulously formalized within the main body of the text to ensure clarity and facilitate comprehensive understanding. Additionally, we provide detailed discussions of implementation in Appendices A.4 to A.6, encompassing dataset information, baselines, experimental settings, and model configurations. The code and pre-trained model weights have been publicized.

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

# A  APPENDIX

## A.1  DETAILED DERIVATION OF THE PLANNING MODULE

In this section, we provide a detailed derivation of the planning module. By approximating the $Q(z)$ with $Q(z|a, q, \mathcal{G})$, the KL divergence can be calculated as

$$
\begin{aligned}
\mathcal{L}_{\text{plan}} = D_{\text{KL}}(Q(z)\|P_\theta(z|q)) &= D_{\text{KL}}(Q(z|a, q, \mathcal{G})\|P_\theta(z|q)), \\
&= \mathbb{E}_{z \sim Q(z|a,q,\mathcal{G})}[\log Q(z|a, q, \mathcal{G}) - \log P_\theta(z|q)], \\
&= -\mathbb{E}_{z \sim Q(z|a,q,\mathcal{G})} \log P_\theta(z|q) + \text{CONST},
\end{aligned}
\tag{11}
$$

where the expectations cannot be computed exactly because of the large number of valid relation paths $\mathcal{Z}$, so we approximate it by using the shortest paths $z \in \mathcal{Z}^* \subset \mathcal{Z}$ between $e_q$ and $e_a$ in KGs (Zhang et al., 2022). This can be formulated as

$$
\mathcal{L}_{\text{plan}} = - \sum_{z \in \mathcal{Z}^*} Q(z|a, q, \mathcal{G}) \log P_\theta(z|q) + \text{CONST}.
\tag{12}
$$

Based on equation 3, by assuming a uniform distribution over the set of shortest paths $\mathcal{Z}^*$, we can rewrite the equation 12 as

$$
\mathcal{L}_{\text{plan}} = -\frac{1}{|\mathcal{Z}^*|} \sum_{z \in \mathcal{Z}^*} \log P_\theta(z|q) + \text{CONST},
\tag{13}
$$

where the CONST is omitted in the final optimization since it makes no contributions to the loss.

## A.2  DETAILED RELATED WORK

### A.2.1  LLM REASONING PROMPT

Many studies have been proposed to harness the reasoning ability of LLMs to handle complex tasks through prompting (Wei et al., 2022; Wang et al., 2022; Yao et al., 2023; Besta et al., 2023; Pan et al., 2023; Zheng et al., 2023d; Jin et al., 2024a;b). Chain-of-Thought (Wei et al., 2022) enables LLMs to generate a reasoning chain that could be helpful to reasoning. Tree of thoughts (Yao et al., 2023) expands the reasoning chain to a tree structure to explore more reasoning paths. Graph of thoughts (Besta et al., 2023) further models the reasoning chain as a graph with an aggregation operation to synergize the reasoning paths. Plan-and-solve (Wang et al., 2023c) prompts LLMs to generate a plan and execute based on it. DecomP (He et al., 2021) prompts LLMs to decompose the reasoning task into a series of sub-tasks and solve them step by step. However, the problem of hallucinations and lack of knowledge affect the faithfulness of the reasoning. ReACT (Yao et al., 2022) treats LLMs as agents, which interact with the environment to get the latest knowledge for reasoning. To explore faithful reasoning, Entailer (Tafjord et al., 2022) introduces a verifier to validate the reasoning steps generated by LLMs. Creswell & Shanahan (2022) present a framework including two LLMs that are used for selecting and generating reasoning steps, respectively. FAME (Hong et al., 2023) introduces the Monte-Carlo planning to generate faithful reasoning steps. RR (He et al., 2022) and KD-CoT Wang et al. (2023b) aim to retrieve relevant knowledge from KGs to produce faithful reasoning plans for LLMs. Think-on-Graph (Sun et al., 2024) and KG-Agent (Jiang et al., 2024) treat LLMs as agents to interact with KGs via prompting to get the latest knowledge for reasoning.

### A.2.2 KNOWLEDGE GRAPH QUESTION ANSWERING

Knowledge graphs contain abundant factual knowledge in a structured format, which has attracted great attention from researchers (Zheng et al., 2022a; 2023a; Pan et al., 2023; Liang et al., 2023). Knowledge graph reasoning aims to derive new insights based on the existing graph structure and facts (Liang et al., 2022; Wang et al., 2023a; Luo et al., 2023b; Zhao et al., 2023). Graph neural networks that effectively capture the structure information have also been widely used in KG reasoning Zheng et al. (2022b; 2023e;b;c); Liu et al. (2023b;a). As a typical reasoning task, knowledge graph question answering (KGQA) aims to obtain answers based on knowledge from KGs, which can be generally divided into three categories: 1) embedding-based methods, 2) retrieval-augmented methods, and 3) semantic parsing methods.

**Embedding-based methods** model the entities and relations in embedding space and design special model architectures to reason answers. KV-Mem (Miller et al., 2016) adopts a Key-Value memory network to store triples for reasoning. EmbedKGQA (Saxena et al., 2020) and NSM (He et al., 2021) utilize the sequential model to mimic the multi-hop reasoning process. QA-GNN (Yasunaga et al., 2021) and Greaselm (Zhang et al., 2021) further adopt the graph neural network to capture the graph structure for reasoning. However, these methods need to design different model architectures, which are not flexible and generalizable.

**Retrieval-augmented methods** aims to retrieve the relative facts from the KGs to improve the reasoning performance. Early works adopt the page rank or random walk algorithm to retrieve subgraphs from KGs for reasoning (Sun et al., 2018; 2019). However, they ignore the semantic information in questions and lead to noisy retrieval results. Zhang et al. (2022) proposes a relation paths-based subgraph retrieval, resulting a better retrieval and QA performance. Other lines of studies retrieving triples from KGs via BM25 (Li et al., 2023) or DPR (Karpukhin et al., 2020; Yu et al., 2022b) to improve the performance of LLMs. They discard the structure information in KGs which leads to suboptimal results. Recently, UniKGQA (Jiang et al., 2022) unifies the graph retrieval and reasoning process into a single model with LLMs, which achieves state-of-the-art performance on KGQA tasks.

**Semantic parsing methods** parse the question into a structural query (e.g., SPARQL) which can be executed by a query engine to get answers (Sun et al., 2020; Lan & Jiang, 2020). ArcaneQA (Gu & Su, 2022) dynamically generates the query based on results from previous steps. RnG-KBQA (Ye et al., 2022) first enumerate all possible queries and then rank them to get the final output. These methods heavily rely on the quality of generated queries. If the query is not executable, no answers will be generated. DECAF (Yu et al., 2022a) combines semantic parsing and LLMs reasoning to jointly generate answers, which also reach salient performance on KGQA tasks.

### A.3 RETRIEVAL ALGORITHM

Given a question $q$ and a relation path as plan $z$, we adopt a constrained breadth-first search to retrieve the reasoning paths. The pseudocode code is presented in Algorithm 1.

We first initialize a queue of current reasoning paths $\mathcal{Q}$ with the entities in the question $\mathcal{T}_q$ (line 3-5). Then, we iteratively expand each reasoning path in $\mathcal{Q}$ by adding the triples that are connected to the entities in the queue following the relation in relation path (line 11-19). The reasoning path is expanded until the length is equal to the length of the relation path. The expanded reasoning path is added to the set $\mathcal{W}_z$ as the final results (line 8-10).

### A.4 DATASETS

We adopt two benchmark KGQA datasets: WebQuestionSP (WebQSP)[4] (Yih et al., 2016) and Complex WebQuestions (CWQ)[5] (Talmor & Berant, 2018) in this work. We follow previous works (Sun et al., 2018; Jiang et al., 2022) to use the same train and test splits for fair comparison. The statistic of the datasets are given in Table 6. The statistics of the answer numbers and reasoning hops are presented in Table 7 and Table 8, respectively.

---

[4] https://www.microsoft.com/en-us/download/details.aspx?id=52763
[5] https://www.tau-nlp.sites.tau.ac.il/compwebq

---

**Algorithm 1:** Retrieve reasoning paths based on relation paths

---

**Input:** Question $q$, relation path $z = \{r_1, r_2, \ldots, r_l\}$, KG $\mathcal{G}$.
**Output:** Reasoning paths $\mathcal{W}_z$.

1  $\mathcal{W}_z \leftarrow \emptyset$;
2  $\mathcal{Q} \leftarrow$ Queue();
3  **foreach** $e_q \in \mathcal{T}_q$ **do**
4  | $\mathcal{Q}$.append($(e_q, [])$); // Initialize queue with question entities.
5  **end**
6  **while** $\mathcal{Q} \neq \emptyset$ **do**
7  | $(s, w_z) \leftarrow \mathcal{Q}$.pop();
8  | **if** $len(w_z) = len(z)$ **then**
9  | | $\mathcal{W}_z$.append($w_z$);
10 | **end**
11 | **if** $len(w_z) < len(z)$ **then**
12 | | $r \leftarrow z[\text{len}(w_z) + 1]$; // Get relation for next step.
13 | | **foreach** $(s, r', t) \in \mathcal{G}$ **do**
14 | | | **if** $r' = r$ **then**
15 | | | | $w'_z$.append($(s, r, t)$); // Expand the reasoning path.
16 | | | | $\mathcal{Q}$.append($(t, w'_z)$);
17 | | | **end**
18 | | **end**
19 | **end**
20 **end**
21 **return** $\mathcal{W}_z$.;

---

Table 6: Statistics of datasets.

| Datasets | #Train | #Test | Max #hop |
|----------|--------|-------|----------|
| WebQSP   | 2,826  | 1,628 | 2        |
| CWQ      | 27,639 | 3,531 | 4        |

Table 7: Statistics of the number of answers for questions in WebQSP and CWQ.

| Dataset | #Ans = 1 | $2 \geq$ #Ans $\leq 4$ | $5 \geq$ #Ans $\leq 9$ | #Ans $\geq 10$ |
|---------|----------|------------------------|------------------------|----------------|
| WebQSP  | 51.2%    | 27.4%                  | 8.3%                   | 12.1%          |
| CWQ     | 70.6%    | 19.4%                  | 6%                     | 4%             |

Table 8: Statistics of the question hops in WebQSP and CWQ.

| Dataset | 1 hop    | 2 hop  | $\geq 3$ hop |
|---------|----------|--------|--------------|
| WebQSP  | 65.49 %  | 34.51% | 0.00%        |
| CWQ     | 40.91 %  | 38.34% | 20.75%       |

Table 9: Statistics of MetaQA-3hop datasets.

| Datasets    | #Train | #Test  | #hop |
|-------------|--------|--------|------|
| MetaQA-3hop | 1,000  | 1,4274 | 3    |

Table 10: Statistics of constructed knowledge graphs.

| KG | #Entities | #Relations | #Triples |
|---|---|---|---|
| Freebase | 2,566,291 | 7,058 | 8,309,195 |
| Wiki-Movie | 43,234 | 9 | 133,582 |

Table 11: The statistics of the instances in the instruction-tuning datasets.

| Planing Data | Retrieval-reasoning Data | Interpretability Data |
|---|---|---|
| 216,006 | 30,465 | 2,000 |

To evaluate the transferability of RoG to other KGs. We further select the MetaQA-3hop dataset (Zhang et al., 2018) which is based on Wiki-Movies KGs[6]. We select 1000 samples from the training split. The statistic of the dataset is presented in Table 9.

Both WebQSP and CWQ can be reasoned based on Freebase KGs[7] (Bollacker et al., 2008). To reduce the size of KGs, following previous works (He et al., 2021; Jiang et al., 2022), we construct a subgraph of Freebase by extracting all triples that contain within the max reasoning hops of question entities in WebQSP and CWQ. Similarly, we construct a subgraph of Wiki-Movies KGs for MetaQA-3hop. The statistics of constructed KGs are presented in Table 10.

We construct the instruction-tuning dataset using the training split of WebQSP and CWQ datasets. For planning optimization, we generate data by extracting the shortest paths that connect the questions and answers, which serve as supervisory signals. In the retrieval-reasoning optimization phase, we feed both extracted shortest paths together with the questions to predict the answers. To enhance interpretability, we randomly select 1000 samples from each dataset along with their reasoning paths and answers. These are then input into ChatGPT to produce interpretive responses, which help empower our method in generating results with good explainability. The statistics of final training datasets are shown in Table 11.

## A.5 BASELINES

We compare RoG with 21 baselines grouping into 5 categories: 1) *Embedding-based methods*, 2) *Retrieval-augmented methods*, 3) *Semantic parsing methods*, 4) *LLMs*, and 5) *LLMs+KGs methods*. The details of each baseline are described as follows.

**Embedding-based methods.**

- KV-Mem (Miller et al., 2016) adopts a Key-Value memory network to store triples and perform multi-hop reasoning by iterative operating on the memory.
- EmbedKGQA (Saxena et al., 2020) models the reasoning on KGs as a sequential link prediction problem by using the embedding of entities and questions.
- NSM (He et al., 2021) utilizes the sequential model to mimic the multi-hop reasoning process.
- TransferNet (Shi et al., 2021) adopts a graph neural network to capture the relevance between entities and questions for reasoning.
- KGT5 (Saxena et al., 2022) finetunes a sequence-to-sequence framework on KGs and generates answers based on the input question.

**Retrieval-augmented methods.**

- GraftNet (Sun et al., 2018) retrieves relevant subgraphs from KGs with entity linking.
- PullNet (Sun et al., 2019) trains a retrieval model composed of a LSTM and a graph neural network to retrieve a question-specific subgraph.

---

[6]https://research.fb.com/downloads/babi
[7]https://github.com/microsoft/FastRDFStore

- SR+NSM (Zhang et al., 2022) proposes a relation-path retrieval to retrieve subgraphs for multi-hop reasoning.
- SR+NSM+E2E (Zhang et al., 2022) further adopts an end-to-end training strategy to jointly train the retrieval and reasoning modules of SR+NSM.

**Semantic parsing methods.**

- SPARQL (Sun et al., 2020) presents a novel skeleton grammar to represent the high-level structure of a complex question with language modes.
- QGG (Lan & Jiang, 2020) generates a query graph for a question by simultaneously adding constraints and extending relation paths.
- ArcaneQA (Gu & Su, 2022) dynamically generates the query based on results from previous steps.
- RnG-KBQA (Ye et al., 2022) first enumerates all possible queries and then ranks them to get the final output.

**Large language models (LLMs).**

- Flan-T5 (Chung et al., 2022) is an enhanced version of T5 models that is instruction finetuned on mixture of tasks.
- Alpaca (Taori et al., 2023) is based on LLaMA and finetuned on an instruction-following dataset.
- LLaMA2-Chat (Touvron et al., 2023) is a large language model that is optimized for dialogue purposes.
- ChatGPT[8] is a powerful closed-source LLM that could follow instructions to conduct complex tasks[9].
- ChatGPT+CoT (Wei et al., 2022) uses the Chain-of-thought prompt to improve the reason ability of ChatGPT.

**LLMs+KGs methods.**

- KD-CoT Wang et al. (2023b) retrieves relevant knowledge from KGs to generate faithful reasoning plans for LLMs.
- UniKGQA (Jiang et al., 2022) unifies the graph retrieval and reasoning process into a single model with LLMs, which achieves state-of-the-art performance on KGQA tasks.
- DECAF (Yu et al., 2022a) combines semantic parsing and LLMs reasoning to jointly generate answers, which also reach salient performance on KGQA tasks.

## A.6 IMPLEMENTATION SETTINGS

For RoG, we use LLaMA2-Chat-7B (Touvron et al., 2023) as the LLM backbone, which is instruction finetuned on the training split of WebQSP and CWQ as well as Freebase for 3 epochs. The batch size is set to 4 and the learning rate is set to 2e-5. We use the cosine learning rate scheduler policy with the warmup ratio set to 0.03. The training is conducted on 2 A100-80G GPUs for 38 hours. During inference, we first adopt the LLM to generate top-$K$ relation paths with the highest probability as the plans. Then, we adopt the Algorithm 1 to retrieve reasoning paths, which are fed into the LLM to reason the final answers.

For LLM beelines, we use zero-shot prompting to conduct KGQA, which directly asks LLMs to answer the question. For other baselines, we directly copy their results reported in UniKGQA (Jiang et al., 2022) and DECAF (Yu et al., 2022a) for comparisons.

For combining the planning module of RoG with different LLMs, we use the planning module to generate plans (relation paths), which are executed on KGs to retrieve the reasoning paths. The retrieved paths are fed into different LLMs during inference by utilizing the reasoning prompts template shown in Appendix A.10.

---

[8] https://openai.com/blog/chatgpt

[9] Experiments are conducted with the ChatGPT model released between July. to Sept., 2023.

Table 12: Performance of RoG on MetaQA-3hop.

| Strategies | MetaQA-3hop | |
|---|---|---|
| | Hits@1 | F1 |
| RoG (train from scratch) | 84.81 | 41.32 |
| RoG (transfer from Freebase) | **88.98** | **50.68** |

Table 13: Training time comparison.

| Method | Training on Freebase | Transferring to Wiki-Movies |
|---|---|---|
| RoG | 38 hours | 2 hours |

## A.7 Additional Experiment Results

### A.7.1 Transferability to Other KGs

We evaluate the transferability of RoG to other KGs. We select the MetaQA-3hop dataset (Zhang et al., 2018) which is based on Wiki-Movies KGs. We select 1000 samples from the training split and utilize two training strategies to finetune RoG: 1) *training from scratch*, where we directly train RoG from LLaMA2-Chat with 1000 samples; 2) *transfer from Freebase*, where we conduct a further finetuning based on RoG trained for Freebase. The results are shown in Table 12. From results, we can see that transfer from Freebase achieves better performance than training from scratch, which demonstrates the transferability of RoG to other KGs.

We also compare the training time on Freebase and transferring to Wiki-Movies KGs. From results shown in Table 13, we can see that the training time on Freebase is 38 hours, while the transferring time is only 2 hours. This demonstrates the efficiency of transferring RoG to other KGs.

### A.7.2 Performance with Different Training Data

In our experiment, we finetune RoG jointly on the training set of both WebQSP and CWQ datasets to maximize the ability of RoG for reasoning on Freebase. To fairly compare with other methods (e.g., UniKGQA (Jiang et al., 2022)) that are only trained on single dataset, we provide additional results of the performance of RoG trained on single dataset. From the results shown in Tables 14 and 15, we can see that RoG trained on single dataset still outperforms the STOA baselines (UniKGQA). Besides, we can also find that jointly training RoG on multiple datasets can further improve the performance. In the future, we will try to generate more QA datasets from Freebase to further improved the reasoning ability of RoG.

### A.7.3 Performance Comparison with Different Finetuned LLMs

In Table 1, we report the zero-shot performance of different LLMs. However, RoG is finetuned on the training split of the QA dataset. To make a fair comparison, we further report the performance of the LLMs finetuned on the training split of the QA dataset in Table 16. From the results, we can see that RoG still outperforms the finetuned LLMs.

Table 14: Performance on WebQSP with different training data.

| Method | Training Data | Hits@1 | F1 |
|---|---|---|---|
| UniKGQA | WebQSP | 77.2 | 72,2 |
| RoG | WebQSP | 81.5 | 61.8 |
| RoG | WebQSP+CWQ | 85.7 | 70.8 |

Table 15: Performance on CWQ with different training data.

| Method | Training Data | Hits@1 | F1 |
|---|---|---|---|
| UniKGQA | CWQ | 51.2 | 49.1 |
| RoG | CWQ | 59.1 | 52.9 |
| RoG | WebQSP+CWQ | 62.6 | 56.2 |

Table 16: Performance comparison with different finetuned LLMs (Hits@1).

| Method | WebQSP | CWQ |
|---|---|---|
| Alpaca-7B (Zero Shot) | 51.78 | 27.44 |
| LLaMA2-Chat-7B (Zero Shot) | 64.37 | 34.60 |
| Alpaca-7B (Finetuned) | 74.57 | 55.98 |
| LLaMA2-Chat-7B (Finetuned) | 73.89 | 53.49 |
| RoG | **85.75** | **62.65** |

### A.7.4 RETRIEVAL COSTS

We present the retrieval time and number of retrieved reasoning paths in Figure 4. From results, we can see that the retrieval time increases with the number of top-$K$ relation paths. Therefore, we should select a proper $K$ to balance the retrieval time and the number of retrieved reasoning paths. In experiments, we set $K = 3$.

### A.7.5 PERFORMANCE ON DIFFERENT HOPS

We present the performance of RoG and its variants on different hops of questions in Table 17. From results, we can see that RoG achieves better performance than its variants on different hops of questions, especially questions with more than 3 hops. This demonstrates the importance of relation paths for improving the reasoning performance of LLMs on complex questions.

### A.7.6 PERFORMANCE ON DIFFERENT ANSWER NUMBERS

We also present the performance of RoG and its variants on questions with different numbers of answers in Table 18. From results, we can see that RoG achieves better performance than its variants on questions with different numbers of answers. Specifically, with the number of answer increasing, the performance of RoG w/o planning decreases significantly due to the lack of knowledge from

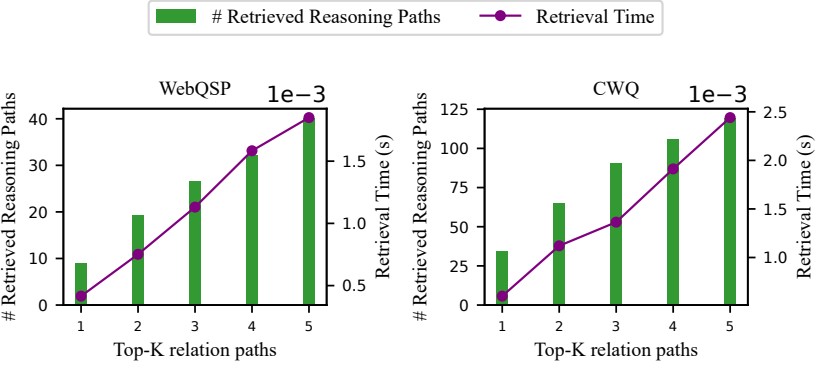

Figure 4: Average retrieval time and average number of retrieved reasoning paths w.r.t. the number of top-$K$ relation paths.

Table 17: F1 scores of RoG and its variants for different hops of questions.

| Methods | WebQSP | | | CWQ | | |
|---|---|---|---|---|---|---|
| | 1 hop | 2 hop | $\geq$ 3 hop | 1 hop | 2 hop | $\geq$ 3 hop |
| RoG | 77.03 | 64.86 | - | 62.88 | 58.46 | 37.82 |
| RoG w/o reasoning | 57.06 | 25.49 | - | 17.06 | 34.25 | 17.07 |
| RoG w/o planning | 50.33 | 51.66 | - | 31.04 | 33.93 | 23.29 |

Table 18: F1 scores of RoG and its variants for questions with varying numbers of answers.

| Methods | WebQSP | | | | CWQ | | | |
|---|---|---|---|---|---|---|---|---|
| | #Ans = 1 | 2 $\geq$ #Ans $\leq$ 4 | 5 $\geq$ #Ans $\leq$ 9 | #Ans $\geq$ 10 | #Ans = 1 | 2 $\geq$ #Ans $\leq$ 4 | 5 $\geq$ #Ans $\leq$ 9 | #Ans $\geq$ 10 |
| RoG | 67.89 | 79.39 | 75.04 | 58.33 | 56.90 | 53.73 | 58.36 | 43.62 |
| RoG w/o reasoning | 33.49 | 52.80 | 58.05 | 66.01 | 16.61 | 27.06 | 40.10 | 34.45 |
| RoG w/o planning | 55.03 | 51.08 | 44.81 | 27.00 | 34.08 | 34.16 | 31.67 | 25.21 |

KGs. Although, RoG w/o reasoning can retrieve more answers to improve the performance. It still is inferior to RoG due to the lack of reasoning ability to remove the noise.

## A.8 CASE STUDIES: RELATION PATHS

We illustrate several examples of relation paths generated by RoG in Table 19.

## A.9 CASE STUDIES: INTERPRETABLE RESULTS

We illustrate several examples of interpretable reasoning results generated by RoG in Table 20.

## A.10 PROMPTS

Planning module aims to generate faithful relation paths as plans for answering the question. The instruction template is presented as follows:

> **Planning Prompt Template**
>
> Please generate a valid relation path that can be helpful for answering the following question: `<Question>`

where `<Question>` indicates the question.

The reasoning module takes the question $q$ and a set of reasoning paths $\mathcal{W}_z$ to generate answers $a$. The instruction template is presented as follows:

> **Reasoning Prompt Template**
>
> Based on the reasoning paths, please answer the given question. Please keep the answer as simple as possible and return all the possible answers as a list.
>
> Reasoning Paths:
> `<Reasoning Paths>`
>
> Question:
> `<Question>`

where `<Reasoning Paths>` denotes the retrieved reasoning paths $\mathcal{W}_z$ which are formatted as a series of structural sentences:

$$e_0 \rightarrow r_1 \rightarrow e_1 \rightarrow \cdots \rightarrow r_l \rightarrow e_l$$
$$\cdots$$
$$e_0 \rightarrow r_1 \rightarrow e_1 \rightarrow \cdots \rightarrow r_l \rightarrow e_l.$$

To exploit the explanation ability of RoG, we design a new instruction template for the reasoning module to generate interpretable results. The instruction template is presented as follows:

---

**Explanation Prompt Template**

Based on the reasoning paths, please answer the given question and explain why.

Here are some examples:
`<Examples>`

Reasoning Paths:
`<Reasoning Paths>`

Question:
`<Question>`

---

where the `Examples` denotes a few-shot human-annotated examples to demonstrate the explanation process.

Table 19: Examples of the generated relation paths.

| Question | Top-3 Relation Paths |
|---|---|
| what does jamaican people speak? | $z_1$ : location.country.languages_spoken
$z_2$ : language.human_language.countries_spoken_in
$z_3$ : location.country.official_language |
| where is jamarcus russell from? | $z_1$ : location.location.people_born_here
$z_2$ : people.person.place_of_birth
$z_3$ : sports.sports_league_draft_pick.player → sports.sports_league_draft_pick.location |
| where did edgar allan poe died? | $z_1$ : people.deceased_person.place_of_death
$z_2$ : people.cause_of_death.people
$z_3$ : people.person.place_of_birth |
| what highschool did harper lee go to? | $z_1$ : people.person.education → education.educational_institution.students_graduates
$z_2$ : education.education.student → education.educational_institution.students_graduates
$z_3$ : people.person.education → education.education.institutio |
| what are the songs that justin bieber wrote? | $z_1$ : music.recording.artist
$z_2$ : music.composition.composer
$z_3$ : music.composer.compositions |
| what are the religions practiced in indonesia? | $z_1$ : people.person.nationality → people.person.religion
$z_2$ : location.statistical_region.religions → location.religion_percentage.religion
$z_3$ : location.country.languages_spoken → religion.religion.languages |
| Lou Seal is the mascot for the team that last won the World Series when? | $z_1$ : sports.mascot.team → sports.sports_championship_event.champion
$z_2$ : sports.mascot.team → sports.sports_team.championships
$z_3$ : sports.sports_championship_event.championship |
| What type of government is used in the country with Northern District? | $z_1$ : location.administrative_division.first_level_division_of → government.form_of_government.countries
$z_2$ : location.administrative_division.first_level_division_of → location.country.form_of_government
$z_3$ : administrative_division.first_level_division_of → government.form_of_government.countries |
| The people from the country that contains Nord-Ouest Department speak what languages today? | $z_1$ : location.administrative_division.first_level_division_of → language.human_language.countries_spoken_in
$z_2$ : location.administrative_division.first_level_division_of → location.country.languages_spoken
$z_3$ : base.aareas.schema.administrative_area.administrative_parent → location.country.languages_spoken |
| What stadium does the team with mascot named Hank play at? | $z_1$ : sports.mascot.team → sports.sports_facility.teams
$z_2$ : sports.sports_team.team_mascot → sports.sports_facility.teams
$z_3$ : sports.mascot.team → sports.sports_team.arena_stadium |
| Which popular sports team in Spain, that won the 2014 Eurocup Finals championship? | $z_1$ : sports.sports_team.championships → sports.sports_team_location.teams
$z_2$ : sports.sports_team.location → sports.sports_team_location.teams
$z_2$ : sports.sports_team.location → sports.sports_team_location.teams
$z_3$ : sports.sports_team_location.teams |
| What educational institution with the mascot named Washington University Bear did Tennessee Williams go to? | $z_1$ : education.educational_institution.mascot
$z_2$ : people.person.nationality → location.location.containedby
$z_3$ : education.education.student → education.education.institution |
| Who is the current head coach of the NFL squad owned by the Rooney family? | $z_1$ : sports.professional_sports_team.owner_s → american_football.football_coach.current_team_head_coached
$z_2$ : sports.professional_sports_team.owner_s → american_football.football_team.current_head_coach
$z_3$ : american_football.football_coach.current_team_head_coached |
| What is the home field of the sports team whose mascot is named Fredbird? | $z_1$ : sports.mascot.team → sports.sports_facility.teams
$z_2$ : sports.sports_team.team_mascot → sports.sports_facility.teams
$z_3$ : sports.mascot.team → sports.sports_team.arena_stadium |

Table 20: Examples of faithful and interpretable reasoning of RoG.

| Question | Lou Seal is the mascot for the team that last won the World Series when? |
|---|---|
| Answer | 2014 |
| Reasoning Paths | Lou Seal → sports.mascot.team → San Francisco Giants → sports.sports_championship_event.champion → 2014 World Series |
| RoG | Based on the provided reasoning paths, we can determine when the team for which Lou Seal is the mascot last won the World Series: 

 1. Lou Seal is associated with the San Francisco Giants. 
 2. The San Francisco Giants are associated with winning the 2014 World Series. 

 Therefore, the San Francisco Giants, the team for which Lou Seal is the mascot, last won the World Series in 2014. |
| Question | what is my timezone in louisiana? |
| Answer | Central Time Zone |
| Reasoning Paths | Louisiana → location.location.partially_containedby → Tangipahoa River → geography.river.origin → Osyka → location.location.time_zones → Central Time Zone 

 Louisiana → location.location.containedby → Alexandria → location.location.containedby → United States of America → location.location.time_zones → Central Time Zone |
| RoG | Based on the reasoning paths provided, it can be determined that Louisiana is in the Central Time Zone. 

 Here's the explanation: 

 1. The first reasoning path tells us that Louisiana is partially contained by the Tangipahoa River, which is a geographical feature. However, it doesn't directly mention the time zone. 

 2. The second reasoning path establishes that Louisiana is contained by Alexandria, which is in turn contained by the United States of America. The United States of America is in the Central Time Zone. 

 Therefore, based on these reasoning paths, the time zone in Louisiana is the Central Time Zone. |
| Question | Which child of Walt Disney died from lung cancer? |
| Answer | Sharon Mae Disney |
| Reasoning Paths | Lung cancer → people.deceased_person.cause_of_death → Sharon Mae Disney 

 Lung cancer → people.deceased_person.cause_of_death → Walt Disney 

 Walt Disney → people.deceased_person.cause_of_death → Lung cancer 

 Walt Disney → people.person.children → Diane Disney Miller 

 Walt Disney → people.cause_of_death.people → Circulatory collapse 

 Walt Disney → people.person.parents → Flora Call Disney 

 Walt Disney → people.person.parents → Elias Disney 

 Walt Disney → people.person.parents → Sharon Mae Disney |
| RoG | Based on the reasoning paths provided, it appears that Sharon Mae Disney is the child of Walt Disney who died from lung cancer. 

 The reasoning path "Walt Disney → people.deceased_person.cause_of_death → Lung cancer" directly connects Walt Disney to lung cancer as a cause of death, and since Sharon Mae Disney is also linked to lung cancer as a cause of death through the path "Lung cancer → people.deceased_person.cause_of_death → Sharon Mae Disney," it can be concluded that Sharon Mae Disney is the child of Walt Disney who died from lung cancer. |

