# OpenReview forum: "Reasoning on Graphs: Faithful and Interpretable Large Language Model Reasoning"
_ICLR.cc/2024/Conference — ICLR 2024 poster_

### Official Review · Reviewer_cFys · 2023-10-27

**Soundness:** 3 good
**Presentation:** 3 good
**Contribution:** 3 good
**Rating:** 8
**Confidence:** 4

**Summary:**

This paper proposes a new framework using external KG to enhance the reasoning ability of LM. The authors finetune LMs (e.g., LLaMA) as the planning and retrieval modules for performing reasoning tasks.

**Strengths:**

Generally, this paper explores an interesting topic with a reasonable method design. Experiment results also look good.

1. Framework design. This reasoning framework design is reasonable: the explicit KG usage makes the reasoning process interpretable and controllable.

2. The two modules (planning and retrieval) finetuning works well. Experimental results show that indeed using the RoG framework with the two tuned modules, better reasoning performance can be achieved.

**Weaknesses:**

However, I have several concerns about this work.

1. From equation 1, I understand the authors want to decompose the prediction task into two parts: getting hidden variable z and then predicting the answer with the hidden variable. However, it's not clear whether equations 2-6 are necessary. Could you provide more intuition behind for explanation?

      Even if motivation exists, I'm not sure why equation 3 holds. Why is there only one path / relation path and it's faithful? For example, there could be multiple solutions/paths; if so, equation 3 might not hold. The authors should provide more justification for the equation 3.

2. One of the key points in the framework is the finetuned LLM (i.e., \theta). However, in the main paper, it's not clear how the LLM is finetuned. It seems the objective functions follow the equation 7. However, these two modules are not evaluated individually and there is no validation loss provided. Only the final framework performance can prove that these modules work as expected, which is not sufficient for readers to know why they can work.

3. Over-claim sentences. The authors claim that their RoG framework can address the hallucination issue and lack of knowledge issue. However, studies with several cases are definitely not sufficient to prove them. I would suggest adding more comprehensive results or changing the claim.

**Questions:**

1. For the KG, is that constructed by yourself and used for all tasks, or is it provided in the dataset?

2. In equation 4, the final equality, the constant term is missing.

3. In Table 2, RoG is finetuned LM, but it's compared with LLMs under the zero/few-shot setting. Is that a fair comparison?

---

> ### Author Response · Authors · 2023-11-15
> **Response to Reviewer cFys**
>
> ## Weakness 1
> > It's not clear whether equations 2-6 are necessary. Could you provide more intuition behind for explanation?
> > There could be multiple solutions/paths; if so, equation 3 might not hold. The authors should provide more justification for the equation 3.
>
> ### Response
> We thank the reviewer for the question and for pointing out the issue in Equation 3.
>
> In equation 1, we want to illustrate the general purpose of RoG, which first generates a set of plans (planning module) and then conducts the reasoning on KGs with the plans (retrieval-reasoning module). However, directly optimizing Equation 1 is intractable due to the combination of hidden variable $z$ [1]. Therefore, we derive the evidence lower bound (ELBO) of Equation 1 in Equation 2-6 and come up with the viable optimization objective in Equation 7.
>
> To clarify the misunderstanding, we want to correct Equation 3 in our paper as follows:
> $$
> \begin{equation}
>     Q(z) \simeq Q(z|a,q,\mathcal{G}) = \begin{cases}
>     & \frac{1}{|\mathcal{Z}|}, \exists w_z(e_q,e_a)\in \mathcal{G},\\\\
>     & 0, \text{else},
>     \end{cases}
> \end{equation}
> $$
> where we assume there are multiple plans $z\in\mathcal{Z}$ which obey a uniform distribution [1].
>
> [1] Singh, D., Reddy, S., Hamilton, W., Dyer, C., & Yogatama, D. (2021). End-to-end training of multi-document reader and retriever for open-domain question answering. NeurIPS 2021, 34, 25968-25981.
>
> ## Weakness 2
> > One of the key points in the framework is the finetuned LLM (i.e., \theta). However, in the main paper, it's not clear how the LLM is finetuned. These two modules are not evaluated individually.
>
> ### Response
> We thank the reviewer for the question. We have revised the paper to clarify the training process of RoG in equation 7 (based on the comments of Reviewer 1bmQ). The objective in equation 7 contains two terms: retrieval-reasoning optimization and planning optimization.  We optimize the objective in Equation 7 by maximizing the likelihood of predicting the correct answer and generating valid plans.
>
> To demonstrate the performance of each module individually, we conduct ablation studies in Table 3, where we remove the planning and reasoning modules, respectively. From the results, we can see that both these modules are helpful for the performance of RoG.
>
> Ablation studies of RoG.
>
> | Method            | WebQSP (F1) | CWQ (F1) |
> | ----------------- | ----------- | -------- |
> | RoG               | 70.81       | 56.17    |
> | RoG w/o planning  | 49.69       | 33.76    |
> | RoG w/o reasoning | 49.56       | 22.26    |
>
> ## Weakness 3
> > Over-claim sentences. The authors claim that their RoG framework can address the hallucination issue and lack of knowledge issue. However, studies with several cases are definitely not sufficient to prove them. I would suggest adding more comprehensive results or changing the claim.
>
> ### Response
> Thank you for the comments. We have revised the paper by changing the claim to "RoG can **alleviate** the hallucination issue and lack of knowledge issue".
>
>
> ## Question 1
> > For the KG, is that constructed by yourself and used for all tasks, or is it provided in the dataset?
>
> ### Response
> We use the KGs preprocessed by previous works [2].
>
> [2] He, G., Lan, Y., Jiang, J., Zhao, W. X., & Wen, J. R. (2021, March). Improving multi-hop knowledge base question answering by learning intermediate supervision signals. WSDM 2021
>
> ## Question 2
> > In equation 4, the final equality, the constant term is missing.
>
> ### Response
> Thanks for pointing it out, we have revised Equation 4. We omit the const in the final objective since it makes no contributions to the optimization. The detailed derivation can be found in Appendix A.1 and in our response to Reviewer 1bmQ.
> $$
> \begin{align}
>     \mathcal{L}\_{\text{plan}} = D\_{KL}(Q(z)||P\_\theta(z|q)) &= D\_{KL}(Q(z|a, q, \mathcal{G})||P\_\theta(z|q)), \\\\
>     &\simeq - \frac{1}{|\mathcal{Z^*}|}\sum\_{z\in\mathcal{Z^*}} \log P\_\theta(z|q),
> \end{align}
> $$
>
> ## Question 3
> > In Table 2, RoG is finetuned LM, but it's compared with LLMs under the zero/few-shot setting. Is that a fair comparison?
>
> ### Response
> Thanks for the questions, we have revised our paper by providing the performance of LLM finetuned on the training set of WebQSP and CWQ. From the results, we can see that RoG still outperforms the LLMs finetuned on the training set of WebQSP and CWQ.  We have revised the paper by providing the results in Appendix A.7.3.
>
> Performance comparison with various finetuned LLMs.
>
> | Method                       | WebQSP (Hits@1) | CWQ (Hits@1) |
> | ---------------------------- | --------------- | ------------ |
> | Alpaca-7B  (Zero Shot)         | 51.78           | 27.44        |
> | LLaMA2-Chat-7B  (Zero Shot) | 64.37           | 34.60        |
> | Alpaca-7B (Finetuned)           | 74.57           | 55.98        |
> | LLaMA2-Chat-7B  (Finetuned)  | 73.89           | 53.49        |
> | RoG                          | 85.75           | 62.65        |

---

> > ### Comment · Reviewer_cFys · 2023-11-16
> > **Response to the reply**
> >
> > Thanks for the detailed explanation! My concerns are resolved, and I'll raise my score to accept. Please update the paper correspondingly.

---

> > > ### Author Response · Authors · 2023-11-16
> > > **Sincere Gratitude from Authors**
> > >
> > > We are thrilled that our responses have properly addressed your concerns! We have accordingly made the necessary update to the paper. We sincerely thank the reviewer for the time and inspiring comments!

---

### Official Review · Reviewer_1bmQ · 2023-10-29

**Soundness:** 3 good
**Presentation:** 2 fair
**Contribution:** 2 fair
**Rating:** 6
**Confidence:** 3

**Summary:**

The authors propose a new knowledge graph retrieval-based fine-tuning algorithm of LLM, RoG, which shows significant improvement over many baselines, including chatGPT, on two KGQA datasets. The method has two training objectives, one retrieval objective, and one planning objective. The LLM is trained to first generate several reasoning paths and then verify and select the best paths based on a KG. The ablation study shows that both objectives are crucial. The fine-tuned LLM can be regarded as a stand-alone planning module for other LLMs, like ChatGPT, and improve their performance.

**Strengths:**

1. The empirical performance of the proposed method seems to be pretty strong on the two KGQA datasets, compared to many baselines.

2. The proposed method seems to be able to better combine the reasoning power of both LLM and KG.

**Weaknesses:**

1. There is some nonrigorous math in the paper. e.g. in equation 4, the expectation and Q should not coexist. It's either $\mathbb{E}_Q \log P$ or $\sum_z Q \log P$. In the next line, $z \in Q$ does not make sense as $Q$ is a probability distribution. Also, the equality does not make sense as there is a CONST in equation 4. Also, I don't think it's a good idea to use equality for an approximation. Similar nonregorousness happens in equation 6. The marginalization in equation 10 does not make sense, as the authors are marginalizing over the conditions. The correctness of the final training objective needs to be double-checked.

2. More datasets to showcase the effectiveness of the proposed method would be great, as there are currently only two in the paper. Would the fine-tuned LLM generalize to other QA datasets, in addition to the datasets that it is fine-tuned on?

3. About RoG as a planning module for other LLMs: I understand that the fine-tuned LLM can also be combined with other LLMs, and improve the performance of these not fine-tuned LLMs. However, according to Table 4, even combining with a stronger LLM (e.g. ChatGPT) cannot improve upon the original fine-tuned LLM. I don't see the usefulness of having this sort of integrability.

**Questions:**

1. Is RoG trained on both WebQSP and CWQ at the same time or is it trained separately on these two datasets? I'm not super familiar with the KGQA baselines, but I wonder if all baselines are trained on the same data as RoG. If the baselines are only trained on one of the datasets each time, then it's not fair to compare RoG with them, if RoG is trained on both of them at the same time.

I'm willing to raise my score if my concerns are properly addressed.

---

> ### Author Response · Authors · 2023-11-15
> **Response to Reviewer 1bmQ**
>
> We sincerely thank the reviewer for specific comments. We have provided a detailed response to each comment below. We hope our answers can properly address your concerns.
>
> ## Weakness 1
> > There is some nonrigorous math in the paper.
>
> ### Response
> We truly thank the reviewer for pointing out these issues! We have carefully revised the paper to fix the nonrigorous math equations.
>
> The revised Equation 4 in our paper is shown here.
> $$
> \begin{align}
>     \mathcal{L}\_{\text{plan}} = D\_{KL}(Q(z)||P\_\theta(z|q)) &= D\_{KL}(Q(z|a, q, \mathcal{G})||P\_\theta(z|q)),\tag{1} \\\\
>     &\simeq - \frac{1}{|\mathcal{Z^*}|}\sum\_{z\in\mathcal{Z^*}} \log P\_\theta(z|q),\tag{2}
> \end{align}
> $$
> where $\mathcal{Z}^*\subseteq\mathcal{Z}$ denotes the set of shortest paths between $e_q$ and $e_a$ in KGs, and $\mathcal{Z}$ denotes all valid relation paths in KGs.
>
> Due to the limitation of space, we put the detailed derivation in Appendix A.1, which is also shown as follows.
>
> The posterior distribution $Q(z)$ can be approximated by $Q(z|a, q, \mathcal{G})$, given as
> $$
> \begin{equation}
>     \tag{3}
>     Q(z) \simeq Q(z|a,q,\mathcal{G}) = \begin{cases}
>     & \frac{1}{|\mathcal{Z}|}, \exists w_z(e_q,e_a)\in \mathcal{G},\\\\
>     & 0, \text{else},
>     \end{cases}
> \end{equation}
> $$
> where we assume a uniform distribution over all valid relation paths $\mathcal{Z}$ [1].
>
> Thus, the KL divergence can be calculated as
> $$
> \begin{align}
> \mathcal{L}\_{\text{plan}} = D\_{KL}(Q(z)||P\_\theta(z|q)) &= D\_{KL}(Q(z|a, q, \mathcal{G})||P\_\theta(z|q)), \tag{4}\\\\
> &=\mathbb{E}\_{z\sim Q(z|a, q, \mathcal{G})}[\log Q(z|a, q, \mathcal{G})-\log P\_\theta(z|q)], \tag{5}\\\\
> &= - \mathbb{E}\_{z\sim Q(z|a, q, \mathcal{G})}\log P\_\theta(z|q) + \text{CONST}, \tag{6}
> \end{align}
> $$
> where the expectations cannot be computed exactly because of the large number of valid relation paths $\mathcal{Z}$, so we approximate it by using the set of shortest paths $\mathcal{Z}^*\subseteq\mathcal{Z}$ between $e_q$ and $e_a$ in KGs. This can be formulated as
> $$
> \begin{align}
> \mathcal{L}\_{\text{plan}} = - \sum\_{z\in\mathcal{Z^*}} Q(z|a, q, \mathcal{G}) \log P\_\theta(z|q) + \text{CONST}. \tag{7}
> \end{align}
> $$
> By assuming a uniform distribution over the set of shortest paths $\mathcal{Z}^*$, we can rewrite Eq. (7) as
> $$
> \begin{align}
> \mathcal{L}\_{\text{plan}} & = - \frac{1}{|\mathcal{Z^*}|}\sum\_{z\in\mathcal{Z^*}} \log P\_\theta(z|q) + \text{CONST}.\tag{8}
> \end{align}
> $$
> We keep the CONST in the equation. However, we omit it in the final objective since it makes no contributions to the optimization.
>
> The revised Equation 6 in our paper is presented here.
> $$
> \begin{align}
>     \mathcal{L}\_{\text{reason}} &= \mathbb{E}\_{z\sim Q(z|a, q, \mathcal{G})}[\log P\_\theta(a|q,z,\mathcal{G})],\tag{9}\\\\
>     &= \sum\_{z\in\mathcal{Z}^*\_K} \log P\_\theta(a|q,z,\mathcal{G}), \tag{10}\\\\
>     &=  \log P\_\theta(a|q,\mathcal{Z}^*\_{K},\mathcal{G}). \tag{11}
> \end{align}
> $$
> The expectation is approximated by sampling $K$ plans from the $\mathcal{Z^*}$, denoted as $\mathcal{Z}^*_K\subseteq \mathcal{Z}^*$. The conversion from Eq. (9) to Eq. (10) is based on the FiD framework [1], where we can simultaneously use multiple plans for reasoning. The FiD framework is introduced in Equation 5 of our paper, which is also shown here.
>
> The FiD framework assumes each $z\in \mathcal{Z}$ contributes independently. The probability $P_\theta(a|q,\mathcal{Z},\mathcal{G})$ can be approximated as the product of each $P_\theta(a|q,z,\mathcal{G})$, which is formulated as
> $$
> \begin{equation}
>     P\_\theta(a|q,\mathcal{Z},\mathcal{G}) = \prod\_{z\in\mathcal{Z}} P\_\theta(a|q,z,\mathcal{G}). \tag{12}
> \end{equation}
> $$
>
> We also want to clarify the marginalization in Equation 10 of our paper is reasonable based on the FiD framework shown in Eq. (12), which can be derived as
>
> $$
> \begin{align*}
> &\text{Because:} \\\\
> & P\_\theta(a|q,\mathcal{Z},\mathcal{G}) = \prod\_{z\in\mathcal{Z}} P\_\theta(a|q,z,\mathcal{G}), \\\\
> &\Rightarrow \\\\
> &\log P\_\theta(a|q,\mathcal{Z}^*\_K, \mathcal{G}) = \log \sum\_{z\in\mathcal{Z}^*\_K} P\_\theta(a|q,z,\mathcal{G}), \\\\
> &\Rightarrow \\\\
> &\underset{\theta}{\operatorname{arg max}}\log P\_\theta(a|q,\mathcal{Z}^*\_K, \mathcal{G}) = \underset{\theta}{\operatorname{arg max}} \log \sum\_{z\in\mathcal{Z}^*\_K} P\_\theta(a|q,z,\mathcal{G}), \\\\
> &= \underset{\theta}{\operatorname{arg max}} \log \sum\_{z\in\mathcal{Z}^*\_K}\sum\_{w\_z\in\mathcal{W}\_z}\prod\_{t=1}^{|a|}P\_\theta(t_i|t_{<i}, q, w_z), \\\\
> &\text{Q.E.D.}
> \end{align*}
> $$
>
> [1] Singh, D., Reddy, S., Hamilton, W., Dyer, C., & Yogatama, D. (2021). End-to-end training of multi-document reader and retriever for open-domain question answering. Advances in Neural Information Processing Systems, 34, 25968-25981.

---

> > ### Author Response · Authors · 2023-11-15
> > **Following Response to Reviewer 1bmQ**
> >
> > ## Weakness 2
> > > Would the fine-tuned LLM generalize to other QA datasets, in addition to the datasets that it is fine-tuned on?
> >
> > ### Response
> > In Appendix A.7.1, we evaluate the transferability of RoG to other KGs. We select the MetaQA-3hop dataset, which is based on Wiki-Movies KGs. The experiment results show that RoG trained on Freebase KGs can be effectively transferred to Wiki-Movies KGs with only 1000 samples. The performance of transferring is better than training from scratch. This indicates that RoG empowers the reasoning on graph ability via training, which can be transferred to new KGs to improve the performance. We also present the training and transferring time here. The transferring time is much less than the training time.
> >
> > Performance on MetaQA-3hop (F1)
> >
> > |  F1   | Only Train on Wiki-Movies | Transferring from Freebase |
> > | :---: | :-----------------------: | :------------------------: |
> > |  RoG  |           41.32           |           50.68            |
> >
> > Training time comparison.
> >
> > | Time | Training on Freebase | Transferring to Wiki-Movies |
> > | :--: | :------------------: | :-------------------------: |
> > | RoG  |        38 hours        |            2 hours            |
> >
> > ## Weakness 3
> > > I don't see the usefulness of having RoG as a planning module for other LLMs.
> >
> > ### Response
> > The reason why other LLMs, when integrated with the planning module of RoG cannot outperform the full RoG is that these LLMs are not trained to understand the reasoning paths and conduct reasoning, as mentioned in Section 4.2. This also demonstrates the importance of reasoning optimization.
> >
> > However, even without training, This experiment shows that the plans generated by the planning module still exhibit great interpretability and can be used by other LLMs to improve their performance. This further opens the possibility of separating the reasoning and generation modules of RoG in the future. For example, we can use a relatively simple LLM to generate the plans and train a larger and more powerful LLM to conduct reasoning based on the generated plans. This experiment shows the flexibility of RoG in actual practices.
> >
> > ## Question 1
> > > Is RoG trained on both WebQSP and CWQ at the same time or is it trained separately on these two datasets? I'm not super familiar with the KGQA baselines, but I wonder if all baselines are trained on the same data as RoG. If the baselines are only trained on one of the datasets each time, then it's not fair to compare RoG with them, if RoG is trained on both of them at the same time.
> >
> > ### Response
> > We thank the reviewer for the question. We first want to clarify that the results reported in the paper are achieved by RoG jointly trained on the training set of both datasets. The reason is that RoG requires the questions and answers that are linked with entities in KGs for training. Since CWQ and WebQSP are both based on Freebase KGs, we use them together to maximize the ability of RoG for reasoning on Freebase KGs. One of the previous KGQA STOA -- Decaf [2] also jointly trains its model with multiple QA datasets on Freebase, including WebQSP and CWQ.
> >
> > To fairly compare with other methods (e.g., UniKGQA) that are only trained on a single dataset, we provide additional results of the performance of RoG trained on a single dataset. From the results (below), we can see that RoG trained on a single dataset still outperforms the STOA baselines (UniKGQA). Besides, we can also find that jointly training RoG on multiple datasets can further improve the performance. In the future, we will try to generate more QA datasets from Freebase to further improve the reasoning ability of RoG. We have revised the paper by providing the results in Appendix A.7.2.
> >
> > Performance on WebQSP.
> > | Method  | Train Data            | Hits@1 | F1   |
> > | ------- | --------------------- | ------ | ---- |
> > | UniKGQA | WebQSP                | 77.2   | 72.2 |
> > | RoG     | WebQSP                | 81.5   | 67.1 |
> > | ROG     | WebQSP+CWQ            | 85.7   | 70.8 |
> >
> > Performance on CWQ.
> > | Method  | Train Data            | Hits@1 | F1   |
> > | ------- | --------------------- | ------ | ---- |
> > | UniKGQA | CWQ                   | 51.2   | 49.1 |
> > | RoG     | CWQ                   | 59.1   | 52.9 |
> > | ROG     | WebQSP+CWQ            | 62.6   | 56.2 |
> >
> > [2] Yu, D., Zhang, S., Ng, P., Zhu, H., Li, A. H., Wang, J., ... & Xiang, B. (2022, September). DecAF: Joint Decoding of Answers and Logical Forms for Question Answering over Knowledge Bases. ICLR 2023.

---

> > > ### Comment · Reviewer_1bmQ · 2023-11-16
> > >
> > > Thank you for the response. It addresses most of my concerns. I still feel the integrability point is a bit weak. I raised my score to 6.

---

> > > > ### Author Response · Authors · 2023-11-16
> > > > **Sincere Gratitude from Authors**
> > > >
> > > > We are truly delighted that our responses have effectively addressed your concerns. We would like to express our sincerest gratitude once again for taking the time to review our paper and provide us with such detailed and invaluable comments!

---

### Official Review · Reviewer_qFDd · 2023-11-04

**Soundness:** 4 excellent
**Presentation:** 4 excellent
**Contribution:** 4 excellent
**Rating:** 8
**Confidence:** 4

**Summary:**

This paper addresses the increasingly important problem of integrating Large Language Models (LLMs) into a more general support framework that can overcome their shortcomings and limitations using axillary techniques.  A compelling 'Reasoning on Graphs' (RoG) approach is introduced to enhance the reasoning capabilities of LLMs by leveraging the structural information of Knowledge Graphs (KGs).  The RoG concept emphasizes the importance of KGs' relational structures in the reasoning processes. The proposed method consists of a planning-retrieval-reasoning framework that generates relation paths grounded by KGs, which serve as reliable plans for subsequent reasoning tasks. These plans guide the retrieval of valid reasoning paths that facilitate faithful and interpretable reasoning by LLMs. The paper addresses two main issues prevalent in previous methods: the tendency of LLMs to produce hallucinated content and the underutilization of KGs' structural data. RoG is optimized through planning optimization, which distills KG structure into LLMs, and retrieval-reasoning optimization, which enables LLMs to produce accurate, KG-supported conclusions. The paper also situates RoG in the context of existing research, identifying its methodological advancements over semantic parsing and retrieval-augmented reasoning approaches.

**Strengths:**

Originality: This paper presents a solid concept for addressing weaknesses in pure LLM model-driven inference by coupling the LLM with a reasoning system.

Quality:  The concept is sensible, compelling, well described, and thoroughly evaluated.  The breadth of comparison techniques is appreciated.

Clarity:  All aspects of the concept, relationship to existing literature, and experimental evaluation are well described.

Significance:  The application community needs actionable approaches to addressing shortcomings to LLMs, and this paper provides one such compelling example.  This result will likely be impactful to future research and implementations.

**Weaknesses:**

Clarity:  The evaluation against ChatGPT appears to use 3.5-turbo.  Please clarify, including the dates of the evaluations -- the implementation of ChatGPT changes over time.

**Questions:**

1. Can you clarify which version of ChatGPT was used?

---

> ### Author Response · Authors · 2023-11-15
> **Response to Reviewer qFDd**
>
> ## Question 1
> > Can you clarify which version of ChatGPT was used?
>
> ### Response
> We greatly appreciate the reviewer’s recognition of our work. We want to clarify that we use ChatGPT-3.5-turbo in our experiments. Experiments are conducted with models released between July. to Sept., 2023. We have revised the paper (Appendix A.5) to clarify this point.

---

> > ### Comment · Reviewer_qFDd · 2023-11-18
> >
> > I appreciate the author's update on the model information.  I am convinced of the originality and quality of the paper, and appreciate the clarity of presentation.  A rating of 8 is well deserved.

---

> > > ### Author Response · Authors · 2023-11-19
> > > **Sincere Gratitude from Authors**
> > >
> > > We sincerely thank you for the positive feedback! We genuinely appreciate the reviewer's time and inspiring comments!

---

### Official Review · Reviewer_ZzpQ · 2023-11-05

**Soundness:** 4 excellent
**Presentation:** 3 good
**Contribution:** 3 good
**Rating:** 8
**Confidence:** 2

**Summary:**

The paper proposes a new approach for questions answering over knowledge graphs. The idea is to use LLMs for QA while exploiting the information in the KG and reasoning over that to alleviate the issues of lack of knowledge and hallucination of LLMs. The main idea is to tune the LLM to generate the relation path needed for finding the final answer, and then instantiate the paths to the answer by searching in the KG. Then feed the instantiated paths that use the actual entities back to the LLM to find the answers that are more faithful to the path of reasoning and less pruned to the hallucination. The experiments are done over two KGQA benchmarks with up to 4 hops of reasoning.  Multiple LLMs (GPT, T5, LLAma, Alpaca) are used and tested. The results show significant improvements compared to a variety of baselines and existing SOTA.

**Strengths:**

The approach is novel and interesting.
The experiments show strong results and improvements over SOTA.
The paper is well written though the organization of the approach description can be improved.

**Weaknesses:**

--The approach section was hard to read.
    --- More specifically, the order of explanation was a bit hard to follow. Before explaining the optimization, I think explaining the flow of information step-by-step will be helpful when you point to Figure 3 in the beginning. In the optimization part, explaining what kind of ground-truth supervision is used was not very explicit. Using the retrieved paths from the KG as a source of supervision could be made clear earlier in the approach.

--The training approach seems to be very costly.  It needs training and instruction-tuning for the LLMs to generate the relation and KG-specific paths. If we train with a specific KG the results will improve in answering questions from that specific KG --which of course is the scope of this work. However, I am not sure if this helps LLM's QA capability in general and the issues set in front including hallucination and lack of knowledge in general.

**Questions:**

--If I understood correctly when you refer to retrieval and reasoning/planning modules of ROG, those are the outcome of instruction-tuning of a specific large language model. When you discussed the ROG model, it was not clear to me what was the base LLM; Which language model was used and tuned for those results of ROG?  when you combine ROG with other language models in Table 4, which one has been used again in the planning module?

---

> ### Author Response · Authors · 2023-11-15
> **Response to Reviewer ZzpQ**
>
> We thank the reviewer for the positive comments. We have revised the manuscript according to your comments and provided a detailed response to each comment below.  We hope our answers can address your questions.
>
> ## Weakness 1
> >  The approach section was hard to read. --- More specifically, the order of explanation was a bit hard to follow. Before explaining the optimization, I think explaining the flow of information step-by-step will be helpful when you point to Figure 3 in the beginning. In the optimization part, explaining what kind of ground-truth supervision is used was not very explicit. Using the retrieved paths from the KG as a source of supervision could be made clear earlier in the approach.
>
> ### Response
> Thank you for your comment. We have revised the approach section to improve its readability. Regarding the framework, we have introduced the components of our method at the beginning and described them in the caption of Figure 2, where we provide a step-by-step description of our approach.
>
> Additionally, regarding the issue of what ground-truth supervision is, we would like to clarify that we use the shortest paths that connect the questions and answer entities in the KG as the supervision signals. We have revised Equation 4 in our paper to clarify this point. The detailed derivation can be found in Appendix A.1 and our responses to Reviewer 1bmQ.
>
> ## Weakness 2
> > The training approach seems to be very costly. It needs training and instruction-tuning for the LLMs to generate the relation and KG-specific paths. If we train with a specific KG the results will improve in answering questions from that specific KG --which of course is the scope of this work. However, I am not sure if this helps LLM's QA capability in general and the issues set in front including hallucination and lack of knowledge in general.
>
> ### Response
> Admittedly, the training process of our method could be costly. However, we want to emphasize that after training, our method enables the generalizability of transferring to new KGs and QA datasets with only a few steps of finetuning.
>
> In Appendix A.7.1, we evaluate the transferability of RoG to other KGs. We select the MetaQA-3hop dataset, which is based on Wiki-Movies KGs. The experiment results show that RoG trained on Freebase KGs can be effectively transferred to Wiki-Movies KGs with only 1000 samples. The performance of transferring is better than training from scratch. This indicates that RoG empowers the reasoning on graph ability via training, which can be transferred to new KGs to improve the model's performance on this new KG. We also present the training and transferring time below (also in Table 12). As can be seen, the transferring time is much less than the training time.
>
> Performance on MetaQA-3hop (F1)
>
> |  Method   | Only Train on Wiki-Movies | Transferring from Freebase |
> | :---: | :-----------------------: | :------------------------: |
> |  RoG  |           41.32           |           50.68            |
>
> Training time comparison.
> | Time | Training on Freebase | Transferring to Wiki-Movies |
> | :--: | :------------------: | :-------------------------: |
> | RoG  |        38 hours        |            2 hours            |
>
> ## Question 1:
> > If I understood correctly when you refer to retrieval and reasoning/planning modules of ROG, those are the outcome of instruction-tuning of a specific large language model. When you discussed the ROG model, it was not clear to me what was the base LLM; Which language model was used and tuned for those results of ROG? when you combine ROG with other language models in Table 4, which one has been used again in the planning module?
>
> ### Response
> We use LLaMA2-Chat-7B as the LLMs in our experiments. When we combine RoG with other language models, we use the planning module to generate plans (relation paths). The plans are executed on KGs to retrieve the reasoning paths. The retrieved paths are fed into different LLMs during inference using the reasoning prompts template shown in Appendix A.10. This enables them to be combined with RoG without retraining. We have revised the paper by adding more details to the implementation settings in Appendix A.6.

---

> ### Comment · Reviewer_ZzpQ · 2023-11-22
>
> Thanks for the response and clarifying the points. I think apart from writing clarity this is a very good paper. I agree the cost of training is worth the improved generalizability.

---

> > ### Author Response · Authors · 2023-11-22
> > **Sincere Gratitude from Authors**
> >
> > We are truly happy that our responses have successfully addressed your questions. Thank you again for the insightful comments and suggestions!

---

### Author Response · Authors · 2023-11-15
**General response to all reviewers**

We sincerely thank the detailed comments from all reviewers. We have carefully revised the paper according to the comments, and the edits have been highlighted in BLUE.  We also provide a detailed response to each comment below. Here we highlight our major revisions, and respond to each reviewer below. We hope our responses can properly address your concerns.

1. Addressing the comments of Reviewer ZzpQ, we have revised the approach section (Section 4) by providing more explanation to improve its readability.
2. Addressing the comments of Reviewer ZzpQ and qFDd, we have revised the experiment settings by providing more details about the used LLMs (Appendix A.6) and version of ChatGPT (Appendix A.5).
3. Addressing the comments of Reviewers ZzpQ and 1bmQ, we have discussed the transferability of RoG to other KGs and provided the results in Appendix A.7.1.
4. Addressing the comments of Reviewers 1bmQ and cFys, we have carefully revised the math equations to make them more rigorous.
5. Addressing the comments of Reviewers 1bmQ and cFys, we have presented the performance of RoG trained on single datasets and compared it with other finetuned LLMs in Appendix A.7.2 and A.7.3, respectively, to enable a fair comparison.

---

### Public Comment · ~Xiaojuan_Tang1 · 2023-11-30
**Similar to RNNLogic**

Hello!

I recently read your paper and found it very intriguing. I would like to understand more about how your method compares with RNNLogic (https://arxiv.org/pdf/2010.04029.pdf). This paper introduces a rule generator and a reasoning predictor. From my perspective, the rule generator seems analogous to the planner in your paper, while the reasoning predictor appears similar to the retrieve-reasoning module you described. Furthermore, I've noticed that the optimization algorithms used in both approaches are strikingly similar. Could you please elaborate on the distinctions and innovations that set your method apart from RNNLogic?

Thank you for your time and for the insights your paper offers.

---

### Public Comment · ~Yixin_Ji2 · 2024-06-06
**Please update the github link**

Hi!

I recently read this exciting paper and want to reproduce it, but the GitHub link is down. Could authors please share an updated link? Looking forward to your reply.

Thanks!

---

> ### Public Comment · ~LINHAO_LUO1 · 2024-06-06
> **Reply from authors**
>
> Hi Yixin,
>
> Thanks for reaching out! We have noticed this issue and are working with Github to solve this issue. Meanwhile, you can access the code via this link temporally: https://drive.google.com/file/d/1Jczz-7urA7FoIKy7f7tY-odDi1UYdUhe/view?usp=sharing
>
> Best,
> Linhao

---

### Public Comment · ~Guancheng_Zhou1 · 2024-06-16

Hi team, thanks for your efforts!


I'm wondering if it's possible to get access to the statistics codes of `Table 8` in your paper, I carefully checked the codes and I didn't find this part


Thank you for your help!

---

> ### Public Comment · ~LINHAO_LUO1 · 2024-06-17
> **Response from author**
>
> Hi Guancheng,
>
> You can use the `get_truth_paths` function provided in `graph_utils.py` to get the ground-truth paths can calculate the statistics shown in Table 8.
>
> Best,
> Linhao

---

### Public Comment · ~Zengyi_Gao1 · 2024-06-17
**Please upload the MetaQA dataset**

Hi!

I recently read this exciting paper and want to reproduce it, but I can't find the MetaQA dataset at [huggingface](https://huggingface.co/rmanluo). Could you share the MetaQA dataset? Looking forward to your reply.

Thanks!

---

### Meta-Review · Area_Chair_EyMy · 2023-12-06

**Metareview:**

This paper produces a new approach for augmenting LLMs with KGs to enable more guided and structured reasoning. The specific novelty in their method RoG, is the planning-retrieval-reasoning framework, where they first generate relation paths grounded by KGs and then use these plans to retrieve valid reasoning paths from the KGs for LLMs to conduct reasoning. All reviewers agreed this was a strong paper, highlight in particular the strength of the empirical results. However, reviewers felt the paper could have also been clearer on some points.

**Justification For Why Not Higher Score:**

Most reviewers agreed the paper should be accepted based on the strength of its empirical results, but didn't highlight the method as particularly novel.

**Justification For Why Not Lower Score:**

See meta-review

---

### Decision · Program_Chairs · 2024-01-16

Accept (poster)